# Disentangling Generalization in Reinforcement Learning

## Abstract

Generalization in Reinforcement Learning (RL) is usually measured according to concepts from supervised learning. Unlike a supervised learning model however, an RL agent must generalize across states, actions and observations from limited reward-based feedback. We propose to measure an RL agent's capacity to generalize by evaluating it in a contextual decision process that combines a tabular environment with observations from a supervised learning dataset. The resulting environment, while simple, necessitates function approximation for state abstraction and provides ground-truth labels for optimal policies and value functions. The ground truth labels provided by our environment enable us to characterize generalization in RL across different axes: state-space, observation-space and action-space. Putting this method to work, we combine the MNIST dataset with various gridworld environments to rigorously evaluate generalization of DQN and QR-DQN in state, observation and action spaces for both online and offline learning. Contrary to previous reports about common regularization methods, we find that dropout does not improve observation generalization. We find, however, that dropout improves action generalization. Our results also corroborate recent findings that QR-DQN is able to generalize to new observations better than DQN in the offline setting. This success does not extend to state generalization, where DQN is able to generalize better than QR-DQN. These findings demonstrate the need for careful consideration of generalization in RL, and we hope that this line of research will continue to shed light on generalization claims in the literature.

## 1 Generalization in Reinforcement Learning

A Reinforcement Learning (RL) agent perpetually finds itself in novel states of an environment. To act intelligently, the agent must generalize its previous experience to new situations. Function approximation helps distill this previous experience into the agent's learnable parameters, which allows previous experience to be leveraged with new state inputs (Boyan and Moore, 1994; Sutton, 1995). While there is a growing literature of new methods for improving generalization of deep RL algorithms, principled and quantitative methods for evaluating generalization remain lacking. This is due in part to the complexity of the MDP problem formulation and the difficulty of disentangling generalization from the performance of RL algorithms in terms of achieving higher expected cumulative reward, *i.e.* return.

One common notion of generalization in RL evaluates an agent's capabilities by checking if it can achieve similar performance in an environment that is similar to, but not exactly the same as, the environment in which it was trained (Whiteson et al., 2011). This can be accomplished through randomization, i.e. by randomizing the parameters underlying the environment, such as wind velocity in helicopter hovering or the mass of objects in a simulator, thereby changing the transition dynamics (Peng et al., 2017). Generalization in this sense draws parallels to supervised learning, where classifiers are often trained on a fixed dataset, and evaluated on a separate testing dataset under the I.I.D. assumption. The agent is said to generalize well if the difference between training error and testing error is small.

While these supervised learning concepts are relevant to RL, there are two problems with taking a cross-environment approach to evaluating RL generalization. First, this paradigm does not disentangle the various aspects of generalization that are required for an RL agent to succeed in its task, whether it be the value estimates or the policy, and whether they are robust to variations in states, observations or actions. This is the question of "what" performance criterion should be measured when we discuss generalization. In RL, we have function approximators for many quantities: state-transition, reward, state-value, action-value and policy. While generalization of state-value is similar to regression, generalization with quantities related to action, such as policy or action-value, do not have supervised learning analogues~~and hence require~~. This is because policies and action-values have as many outputs as actions, and only the action taken by the agent is updated, which necessitates separate consideration.

Second, the paradigm follows practice in supervised learning to impose a strict separation of test and train environments. This practice effectively focuses on transfer performance *across* environments but fails to evaluate the agent's ability to generalize *within* a single environment. I.e. it fails to answer the question of "how" to measure generalization under a performance criterion. Unlike supervised learning, the agent's state distribution changes during learning because the policy changes. Even where we have randomly generated training and testing sets of environments, the environments' complex dynamics do not admit ground-truth labels to determine optimal actions or values for comparison and evaluation. These environments only allow an agent's generalization capabilities to be measured in terms of Monte-Carlo rollouts of the learned policy, leaving us restricted to the single performance criterion of return and preventing us from specifically measuring important nuanced differences in the agent's ability to generalize. In addition, because the upper bound on return is unknown to the researcher, informed judgments about the quality of the policy is very difficult to make.

Within the RL-oriented literature on generalization, there are two distinct categories of research. The first proposes environments and methodologies for measuring RL generalization. Examples of this approach are ProcGen/CoinRun ~~(Cobbe et al., 2020; ?)~~ (Cobbe et al., 2020; 2019), randomized-reward CartPole (Zhang et al., 2018a), the Grid-World maze (Zhang et al., 2018b), observation projection ~~(?)~~ (Song et al., 2020) and the hierarchy of state generalization (Witty et al., 2018). These efforts are aimed at the second issue raised above, i.e. "how" generalization should be measured in RL. Underlying these approaches is specifying how to split the environment into testing and training scenarios. Early work by Zhang et al. (2018a) proposes using separate seeds. This, however, does not ensure that the states encountered by the agent are truly separate. Other works, such as those by ~~Cobbe et al. (2020); ?); Elsayed et al. (2020)~~ Cobbe et al. (2020; 2019); Elsayed et al. (2020), procedurally generate separate environments for testing and training. When generalization is measured on truly separate testing and training environments, we are able to determine whether an agent's policy is generalizing from one environment to another. Again, this formulation does not allow us to study generalization within a single environment, nor does it allow us to measure generalization of the value functions. The second category proposes or investigates RL methods that improve generalization. These include regularization experiments in Atari (Farebrother et al., 2018) and continuous control ~~(?)~~ (Liu et al., 2021), contrastive similarity embeddings (Agarwal et al., 2021) and bisimulation metrics ~~(?)~~ (Zhang et al., 2021). There is also the hypothesis that better, and hence more generalizable, representations arise from auxiliary tasks ~~(?)~~ (Jaderberg et al., 2017), which is also suspected to be the reason for the success of distributional RL ~~(?Bellemare et al., 2017)~~ (Dabney et al., 2018; Bellemare et al., 2017) and has recently been investigated in the offline setting ~~(?).~~ (Agarwal et al., 2020).

~~Previous work closest to ours is using Contextual Decision Processes (CDPs) as~~ To understand generalization in RL, we use the Contextual Decision Process (CDP) framework, which is a problem class for theoretical analysis of RL algorithms that use function approximation (Du et al., 2019; Jiang et al., 2017; Dann et al., 2018). The CDP problem formulation renders the states unobservable, but allows the agent to view observations that contain enough information to recover the state. This formulation enables the study of function approximation as applied to RL and has significant implications for RL in general. For example, ~~it helps to~~ Du et al. (2019) show that there exist algorithms with exponentially more efficient explo-

ration than Q-learning ~~(Du et al., 2019)~~, a result suggesting that RL algorithms should be designed to take advantage of function approximation and its generalization abilities, rather than naively extending tabular algorithms with function approximation. However, no existing work has leveraged CDPs empirically, despite that it connects supervised learning with RL. The ~~MNIST gridworld~~ 2D MNIST maze environment used by ~~?~~ Lee et al. (2019) may be considered a ~~CDP,~~ simple CDP, where the observations are deterministic and equivalent to state, but is not recognized as such. The work by ~~?~~ Song et al. (2020) proposed projecting the state of simple control environment and varying this projection between training and testing sets. These two works have similar goals to ours. However, the first work does not investigate generalization and neither makes use of the ground-truth values to probe generalization rigorously ~~, such as in the label-corruption experiment by (Zhang et al., 2017) that we will extend to RL~~. Our work is uniquely analogous to generalization in supervised learning, simultaneously answering "how" generalization should be measured in RL and "what" should be measured. In answering theses two questions, we disentangle generalization across three axes: states, observations and actions. Finally, this disentangled perspective shows how different generalization mechanisms benefit the different axes of generalization.

## 2 A PROBLEM FORMULATION FOR EVALUATING GENERALIZATION IN RL

To make generalization in RL concrete, we first make clear the Markov Decision Process (MDP) problem formulation that underpins task specification in RL ~~(?Lattimore and Szepesvári, 2020)~~ (White, 2017; Lattimore and Szepesvári, 2020). An MDP $\mathcal{M}$ is defined by the tuple $(\mathcal{S}, \mathcal{A}, r, T, \mu_0, \gamma)$, where $\mathcal{A}$ denotes the action space, $\mathcal{S}$ is the state space, $r : \mathcal{S} \times \mathcal{A} \to \mathbb{R}$ is the reward function that maps a state and an action to a reward, $T : \mathcal{S} \times \mathcal{A} \times \mathcal{S} \to [0, 1]$ is the state transition function, $\mu_0$ is the initial state distribution and $\gamma \in [0, 1]$ is the discount factor. A policy that interacts with the MDP is defined by a mapping from states to a distribution over actions $\pi : \mathcal{S} \to \Delta(\mathcal{A})$, which includes deterministic policies that concentrate the probability mass on a single action. The discounted state visitation distribution for following the policy $\pi$, starting in state $s_0$ and taking action $a_0$, is defined as $d^\pi_{s_0,a_0}(s) = (1-\gamma) \sum_{t=0}^{\infty} \Pr(s_t = s \mid s_0, a_0, a_{t+1} \sim \pi, s_{t+1} \sim T(s_t, a_t))$ (Agarwal et al., 2019). If action $a_0$ is not given, then we define $d^\pi_{s_0}(s) = \mathbb{E}_{a_0 \sim \pi}[d_{s_0,a_0}(s)]$. For any given policy, we define the state-value and action-value functions as expectations over the discounted state visitation distribution[1],

$$v_\pi(s) = \frac{1}{1-\gamma} \mathbb{E}_{s' \sim d^\pi_s, a' \sim \pi}[r(s', a')], \quad q_\pi(s, a) = \frac{1}{1-\gamma} \mathbb{E}_{s' \sim d^\pi_{s,a}, a' \sim \pi}[r(s', a')]. \quad (1)$$

We denote the optimal policy that maximizes the expected cumulative reward (*i.e.* return) by $\pi^* = \arg\max_\pi \mathbb{E}_{s \sim \mu_0}[v_\pi(s)]$. Lastly, we use the shorthand $v^*(s) = v_{\pi^*}(s)$ and $q^*(s, a) = q_{\pi^*}(s, a)$ for the value of the optimal policy (Sutton and Barto, 2018).

### 2.1 THE CURRENT PARADIGM: GENERALIZATION ACROSS MDPS

Studying generalization in RL is challenging in part due to the MDP problem formulation. As an agent interacts and learns in an environment, which is assumed to be an MDP, the agent's policy changes and it continuously visits new states. Due to the changing policy, it is not possible to forbid an agent from entering a state that a practitioner would like to use later for testing. To evaluate RL agents' ability to generalize, one can instead use a generalized environment $\mathcal{E} = (\boldsymbol{\Theta}, \rho)$ that provides a distribution $\rho$ over a collection of environments $\boldsymbol{\Theta} = \{\mathcal{M}_i\}_{i=1}^{N}$ (Whiteson et al., 2011). It is then the responsibility of the environment designer to ensure that there is similarity among the environments $\mathcal{M}_i$ such that generalization is possible. Yet, the environments must also be different enough to avoid inadvertently training on experience that is later tested. The state and action spaces, for example, should not change between the environments. Lastly, one can think of the distribution over environments $\rho$ as a distribution over the distribution of start states $\mu_i$ for each $\mathcal{M}_i$.

---

[1] We use the continuing formulation in Equation 1 for ease of reading. The equation still holds in the episodic setting by replacing the factor of $\frac{1}{1-\gamma}$ by the expected length of the episode $\mathbb{E}_\pi[T]$ (Bojun, 2020). In our experiments, we use the empirical distribution.

Even with generalized environments, such as procedurally generated games (Cobbe et al., 2020), we are unable to probe the details of an agent's generalization ability during learning. By measuring only the Monte-Carlo return of the learned policy on an unseen environment instance, we are restricting the evaluation criterion to on-policy control in this new environment instance. Even if we accept that this is the performance criterion that matters, we can only evaluate the relative performance of the policy because we typically do not know the optimal policy. Lastly, this evaluation protocol cannot evaluate the agent's ability to generalize in a single environment. Harnessing generalization within a single environment is as important as transfer to new environments, and was the original goal for combining reinforcement learning with function approximation (Sutton, 1995; Boyan and Moore, 1994; Tesauro, 1995; Murphy, 2005).

## 2.2 Proposed Paradigm: Evaluating generalization using CDPs

The Contextual Decision Process (CDP) adds an observation space (also referred to as context) $\mathcal{O}$ to the MDP, and renders the state-space unobservable to the agent (Jiang et al., 2017). Observations are provided to the agent by an emission function $\phi : \mathcal{O} \times \mathcal{S} \to [0, 1]$ that samples an observation $o \in \mathcal{O}$ at a state $s \in \mathcal{S}$. The reward and state transitions are still defined as functions of the unobservable states $s \in \mathcal{S}$, and the agent is responsible for learning how to map the observations to states. We write the set of observations associated with a specific state $s$ as $\mathcal{O}_s = \{o : \phi^{-1}(o) = s\}$, for some decoding function $\phi^{-1}$. In this work, we assume that the observation space is disjoint: each state $s \in \mathcal{S}$ is associated with a subset of the observations $\mathcal{O}_s \subset \mathcal{O}$ where $\mathcal{O}_s \cap \mathcal{O}_{s'} = \emptyset$ for $s \neq s'$. The disjoint property ensures the existence of the decoding function $\phi^{-1}$ that maps each observation to a single state. Recent literature introduced the term "Block MDP" to refer to a CDP with the disjoint property (Du et al., 2019). The experiments in this paper only use Block MDPs, but this is to focus on the generalization problem without introducing partial observability.

While CDPs have been the subject of theoretical study, no work has investigated its use as an evaluation platform for studying generalization. To accomplish this, we outline how a CDP can be combined with a supervised learning dataset to allow for generalization across observations. To generate observations, we choose to use images from a supervised learning dataset. Given a finite state MDP, we use a K-way classification dataset $\{(x_i, y_i)\}_{i=1}^n$, with $x_i \in \mathbb{R}^d$ and labels $y_i \in \{1, \dots, K\}$. If the total number of classes $K$ is equal to the number of states $|\mathcal{S}|$ in the MDP, then each state can be uniquely identified with a class label, yielding the observation sets $\mathcal{O}_s = \{x_i : y_i = s\}$. If the states have structure, such as if each state is an xy-coordinate in a gridworld environment $s = (s^{(1)}, s^{(2)})$, then the emission function can concatenate the images corresponding to each component, $\mathcal{O}_s = \{[x_i, x_j] : y_i = s^{(1)}, y_j = s^{(2)}\}$. Using the provided training and testing split of supervised learning datasets, one can then use training data during learning and testing data during evaluation. With this formulation, generalization to new observations can be studied within a single environment. In the next section, we discuss how to extend this idea to states and actions.

## 3 Characterizing Generalization in RL

With the CDP problem formulation, we can now reexamine the differences between generalization in supervised learning and RL. In supervised learning, measuring generalization requires a training distribution $\mathcal{D}_{\text{train}}$, testing distribution $\mathcal{D}_{\text{test}}$ and performance criterion $g$ which is low if the function approximator $f$ is performant. The training and testing set come from the same population distribution, but we maintain and sample from separate training and test sets and denote them by their own distribution. We say that a function approximator $f$ exhibits good generalization if the generalization gap,

$$\mathcal{G}_{SL}(f) = g(f, \mathcal{D}_{\text{test}}) - g(f, \mathcal{D}_{\text{train}}) \tag{2}$$

is small. When the training error can be minimized to zero, as in supervised learning with over-parameterized neural networks, generalization can be characterized by the testing performance alone. As discussed in Section 2.1, generalization in RL is often measured by

the difference in value achieved by a policy $\pi$ over a distribution[2] of training instances $\rho_{train}$ and testing instances $\rho_{test}$ from a generalized environment $\mathcal{E}$,

$$\mathcal{G}_{RL-MDP}(\pi) = \mathbb{E}_{s \sim \rho_{train}}[v_\pi(s)] - \mathbb{E}_{s \sim \rho_{test}}[v_\pi(s)] \qquad (3)$$

where the expectation and the value are both estimated using Monte-Carlo methods. Note that a higher $v_\pi$ is more performant, and so we flip the order in which test and train appear in the gap so that $\mathcal{G} \geq 0$. While Equations 2 and 3 seem otherwise similar, they have a number of differences. First, the performance criterion in RL has no ground-truth label for comparison. An RL agent attempts to maximize the value, but the true optimal value function $v^*$ is unknown in most environments used to study generalization. A randomly initialized function will exhibit similarly poor performance during testing and training and hence the generalization gap will be small. In supervised learning, we can detect that, although the generalization gap is small, the error is high (or performance is low, in the case of value or accuracy). Second, there is a difference in the distributions that are used in the expectation. A supervised learning model learns to produce certain outputs, conditioned on certain inputs. The model is then tested on held out inputs. An RL agent evaluated in the current paradigm produces actions conditioned on states. However, it is then evaluated on entire environments where it chooses actions and produces its own next states as input. Lastly, there is a potential mismatch in the performance criterion. The value of the policy is not the explicit objective being optimized by most reinforcement learning algorithms. This is not an issue in supervised learning because the optimization objectives are calibrated surrogates (Chen et al., 2019).

The two issues that we explicitly address is the discrepancy in the measuring distributions and a lack of ground-truth labels. These two issues make operationalizing the supervised learning notion of generalization difficult in RL. Using the CDP formulation from Section 2, we are able to alleviate these two issues. First, we are able to sample from any distribution over states, observations and actions, meaning that we do not have to explicitly rollout the policy for evaluation. Second, we can use dynamic programming with the unobserved latent states to recover the optimal value and policy, and use those as a comparison to the learned value and policy. One question remains: what performance criterion should we use to test an agent's ability to generalize?

Re-examining the current paradigm for measuring generalization in RL, we see how the CDP approach can help remedy the compromises made when learning in complex environments. An RL agent's task is to maximize return and hence, most RL research reports the returns that an agent receives from interacting with the environment. Due to the the lack of ground-truth labels, the performance criterion $g$ is limited to the Monte-Carlo estimate of $v^\pi$. While Monte-Carlo returns do demonstrate policy performance, it does not probe the agent in its estimates of the actual values being learned, and is only one part of the generalization puzzle. Given the optimal policy and action-values, an agent can be evaluated with more acuity than Monte-Carlo returns. One possibility is to measure the accuracy of the policy, where accuracy is given by the agreement of the learned policy with the true optimal action. Another possibility is to measure the mean squared error between the learned action values and the true optimal action values. While the optimal policy is defined as a function of state, action-values can be evaluated on arbitrary actions. This means that even if an action was not taken in a particular state, we can evaluate the agent's estimate of that particular action-value. Thus, we consider three types of generalization: state, observation, and action generalization. To ease the notation, we define the joint training distribution over state, observation and action, starting at state $s_0$ as $p_{s_0,tr}^{\pi,\pi'}(s,o,a) = d_{s_0,tr}^{\pi}(s)\phi_{tr}(o|s)\pi'_{tr}(a|o)$, with the testing distribution defined similarly. The second policy $\pi'$ is often the same as $\pi$, but this second degree of freedom will be needed for action generalization. Using this shorthand, we define the generalization gap of a function approximator $f$ with respect to a performance criterion $g$ as,

$$\mathcal{G}_{RL-CDP}(f) = \mathbb{E}_{(s',o',a') \sim p_{s,test}^{\pi,\pi'}}[g(s',o',a',f)] - \mathbb{E}_{(s',o',a') \sim p_{s,train}^{\pi,\pi'}}[g(s',o',a',f)]. \qquad (4)$$

---

[2]As described in Section 2.1, $\rho$ is a distribution over the distribution of start states for each environment

In the next two subsections, we will investigate each component of this joint distribution and its connection to generalization across each of the three axes: state, observation and action. We will also show examples of performance criteria $g$ that can be ~~minimized~~measured during training, other than the scaled negative reward $g(s', o', a', \pi) = -\frac{1}{1-\gamma} r(s', a')$, that may be of interest when studying generalization in RL using CDPs. Other criteria are necessary because the scaled negative reward, which defines value under a stationary state distribution, is trajectory dependent and conflates generalization across states, observations and actions.

## 3.1 Observation and State-space Generalization

Observation-space generalization is the most commonly explored notion of generalization in RL (see, i.e. ?Song et al. (2020)). Given an observation $o$, sampled from the test distribution $\phi_{test}$ that the agent has not seen before, the agent is evaluated on quantities related to that observation. In model-free algorithms that learn action-values parameterized by $\theta$, for example, an agent may be evaluated on the mean squared error of its value of the greedy policy, $g_{mse}(o, s, a, \theta) = (v(o; \theta) - v^*(s))^2$. We focus on action-unconditional quantities to separate observation-space generalization from action-space generalization that we will discuss in the next section. Another example of an action unconditional performance criterion is the accuracy of the greedy policy induced by $q$ parameterized by $\theta$, $g_{acc}(o, s, a, \theta) = \mathbb{1}(\arg\max_a q(o, a; \theta) = \pi^*(s))$ where $\mathbb{1}$ is the indicator function.

There is often no distinction between observations and states in the current RL literature because, except in toy problems, the underlying MDP (and hence, the true Markovian state) is not known. However, there are important differences. The state transition and reward are functions of the state, hence generalization across states must necessarily generalize to different state-transition and reward functions. For online algorithms, measuring state generalization is not straightforward because the agent does not put weight on states outside of its state visitation distribution $d_{s_0}^\pi(s)$. Expecting an agent to generalize to an arbitrary state outside of this distribution is similar to the zero-shot generalization problem, and would require a meta-learning approach. More reasonable would be to expect the agent to generalize to states near the distribution $d_{s_0}^\pi(s)$. One way this can be achieved is if the observation $o' \sim \phi_{test}(\cdot|s')$ of a state that has not been encountered by the agent $s' \sim d_{s_0, test}^\pi$ shares some structure with observations $o \sim \phi_{tr}(\cdot|s)$ of seen states $s \sim d_{s_0, test}^\pi$. Then, the function approximator $f$ can generalize to this new state via its observation space generalization. It remains to construct an environment with naturally unreachable states, for which the observation structure permits generalization, such as in certain gridworld setups (Witty et al., 2018).

## 3.2 Action-space Generalization

Generalization to out-of-distribution actions is a less explored notion compared to observation or state-space generalization. To evaluate action-space generalization, the agent ~~sould~~ should be evaluated on quantities related to actions that were not taken or that have low probability under the current policy. In model-free algorithms that learn action-values parameterized by $\theta$ for example, the agent can be evaluated on its estimate of the action-value for an action $a$ not ever taken by the agent in state $s$, and hence not taken in any of it's observations $o \in \mathcal{O}_s$, $g_{act}(s, o, a, \theta) = (q(o, a; \theta) - q^*(s, a))^2$. One way this can be formalized is by setting $\pi'$ to put all of its support on actions outside of the support of $\pi$. This setting is particularly important for offline learning, where the $\pi$ would be the behavior policy and $\pi'$ is the target policy we are trying to learn.

Control is one of RL's main differences from supervised learning, wherein the agent collects the data that it uses to learn. We argue that action-space generalization is more important than observation or state-space generalization because policies in RL tend to place non-zero probability on all actions for exploration purposes. If the agent chooses to explore when faced in an unknown situation, it must leverage its ability to generalize in action-space. It is thus surprising that action-space generalization has seen limited investigation. The work of Chandak et al. (2019); Dulac-Arnold et al. (2015) finds that learned action representations

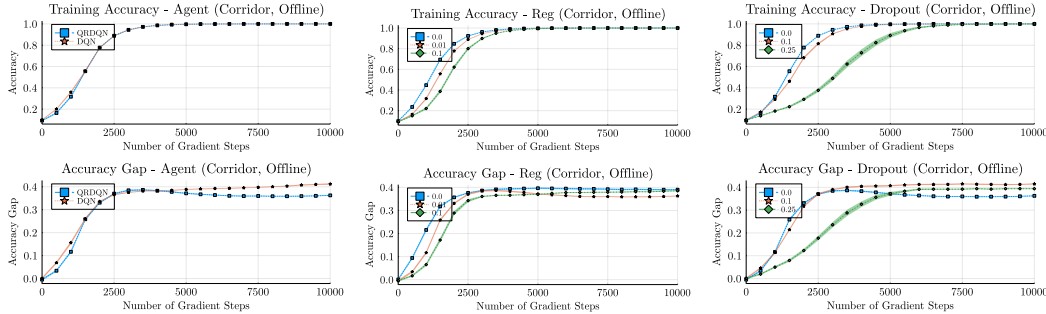

Figure 1: Observation generalization experiment after offline learning with uncorrupted labels in the corridor MNIST CDP. Top: Train accuracy. Bottom: Difference between testing and training accuracy. Left-Right: best agent, regularization and dropout rate respectively.

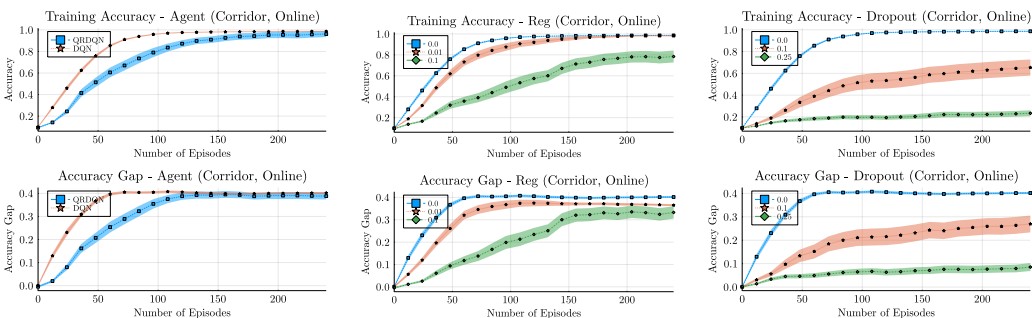

Figure 2: Observation generalization experiment after online learning with uncorrupted labels in the corridor MNIST CDP. Top: Train accuracy. Bottom: Difference between testing and training accuracy. Left-Right: best agent, regularization and dropout rate respectively.

are able to improve the policy's performance in environments with large discrete action spaces. While the authors claim that the learned representation improves performance, there is no specific study of this finding. Furthermore, no work has explicitly measured an agent's ability to generalize to an unseen action.

## 4 EXPERIMENTS

All of our experiments are conducted in a contextual decision process, where the observations are derived from the MNIST dataset (LeCun et al., 2010) and the underlying MDP is either a corridor environment or a four rooms gridworld (Sutton et al., 1999). See the Appendix for details on ~~each environment, including state, action and observation spaces.~~ the experimental protocol, as well as the environments.

While these two environments are relatively simple, they provide a thorough testbed for probing different aspects of generalization for an RL agent. From the agent's perspective, the problem of mapping an observation to a state alone is challenging. Early in training, each observation is unique and so the agent does not know if it is making progress until it reaches the goal. Even after reaching the goal, it is not immediately clear how many unique states were encountered in the trajectory. The agent must ~~do this for each state, and then learn that the~~ learn the mapping from observation to action-value and the fact that action-values ~~in these~~ at different observations are the same if the underlying state is the same. Note that the optimal policy in the corridor environment is the classification policy. However, the ~~classification~~ policy must be learned from a sparse reward signal that occurs only at the end of the episode. Taken together, the agent must generalize across different states, observations and actions from limited experience.

We focus on off-policy algorithms, namely DQN (Mnih et al., 2013) and Quantile-Regression DQN (Dabney et al., 2018), because their ability to learn from arbitrary data necessitates generalization. Our goal for each experiment is to rigorously evaluate generalization across states, observations and actions in different RL training regimes, and with different generalization mechanisms. We can do this because the ~~observations can~~ states, observations and actions can all be partitioned into training, validation and testing sets ~~. In addition to observation generalization, we also investigate~~ in our environment setup. We identify three mechanisms that have been claimed to improve observation generalization in RL: QR-DQN (Agarwal et al., 2020), as well as L2 regularization and dropout (Farebrother et al., 2018; Igl et al., 2019; Cobbe et al., 2019). We begin by rigorously evaluating these previous claims of observation generalization. We then investigate the effects of these generalization mechanisms on action-generalization in the corridor environment and on state-generalization in the four rooms environment. We ~~focus on off-policy algorithms, namely DQN (Mnih et al., 2013) and Quantile-Regression DQN (?), because their ability to learn from arbitrary data necessitates generalization~~ also introduce corrupted variants of these environments, where the class labels in the dataset are shuffled, to test the agent's ability to memorize it's training input. We evaluate ~~both off-policy model-free algorithms~~ all combinations of RL algorithm and generalization mechanism after either online and offline training. In the online setting, the agent will interact with the environment while updating its policy. ~~The agent will be periodically evaluated against ground-truth quantities pertaining to the environment and test set observations.~~ In the offline setting, the agent will learn from a logged set of experience in a replay buffer without any direct interaction with the environment during learning. ~~We also introduce corrupted variants of these environments, where the class labels in the dataset are shuffled to test the agent's ability to memorize it's training input. For all experiment in non-corrupted environments, we experiment with L2 regularization and dropout, as they are common regularization techniques in machine learning and some have reported that these techniques improve generalization.~~

## 4.1 Observation generalization

To measure observation generalization, we train an agent using observations from the training set and evaluate on a held out test-set. We only show the results for the corridor environment here, but have additional experiments on four rooms in the Appendix. Referring to the offline learning experiments depicted in Figure 1, we see that QR-DQN is able to generalize better than DQN, as shown in its lower accuracy generalization gap. We also find that small amounts of regularization indeed lowers the generalization gap, whereas dropout does not improve observation generalization. In the online learning experiments, shown in Figure 2, QR-DQN and DQN both generalize to new observations similarly. Interestingly, the generalization gap is overall larger in the online setting. We also find that setting the regularization strength to 0.1 now inhibits learning. Small amounts of regularization still improves generalization to new observations however. Lastly, we see that dropout hurts learning in the offline setting.

Drawing inspiration from the work of Zhang et al. (2017), we also run an experiment where the labels are randomized to see if DQN and QR-DQN have similar capacity to memorize the data. Referring to Figure 3, we find that DQN and QR-DQN (with a various number of quantile heads) can easily memorize all the training experience in corridor. In the online setting, larger QR-DQN agents begin to struggle to fit the training data. This difficulty also extends to both the offline and online setting in four rooms, where the observations are higher dimensional.

## 4.2 Action generalization

For action generalization, we focus on the corridor environment with offline learning from a behaviour policy that systematically avoids a certain action in each state. We then measure the action value estimate for the state-action pair that is never seen in the data and compare it to the true action-value. Referring to Figure 4, we find QR-DQN is better able to generalize

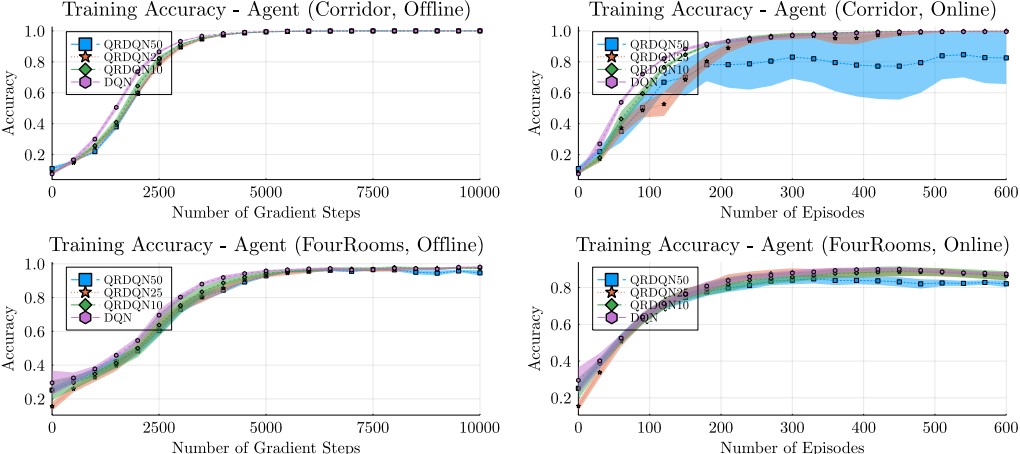

Figure 3: Corruption experiments where all MNIST labels are randomly reassigned to test whether an agent can memorize its experience. Top Left: Corridor Offline. Top right: Corridor online. Bottom Left: Four rooms offline. Bottom right: Four rooms online

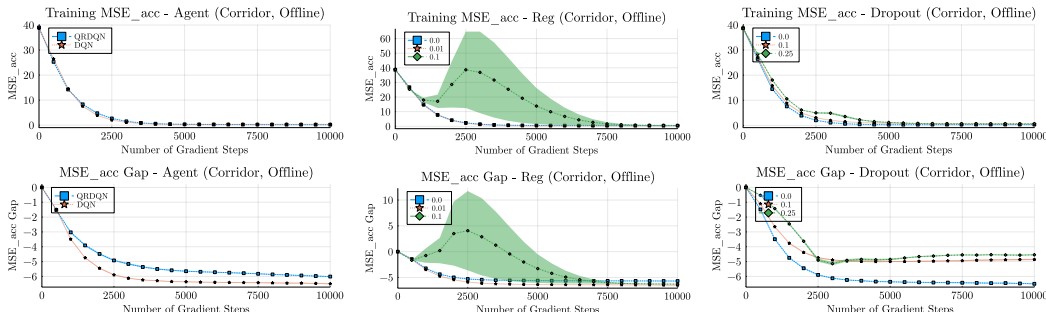

Figure 4: Action generalization with offline learning from a policy that never takes action $a = s - 1$ in state $s > 1$. Top: Average Training MSE for action-value $q(s, s - 1)$. Bottom: Difference between testing and training MSE. Left-Right: best agent, regularization and dropout rate respectively.

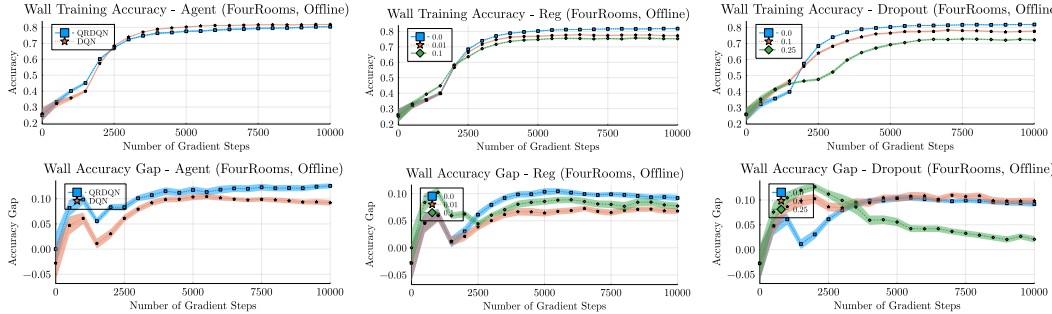

Figure 5: State generalization experiment after offline learning with uncorrupted labels in the four rooms gridworld CDP. Top: Train accuracy. Bottom: Difference between testing and training accuracy on wall states. Left-Right: best agent, regularization and dropout rate respectively.

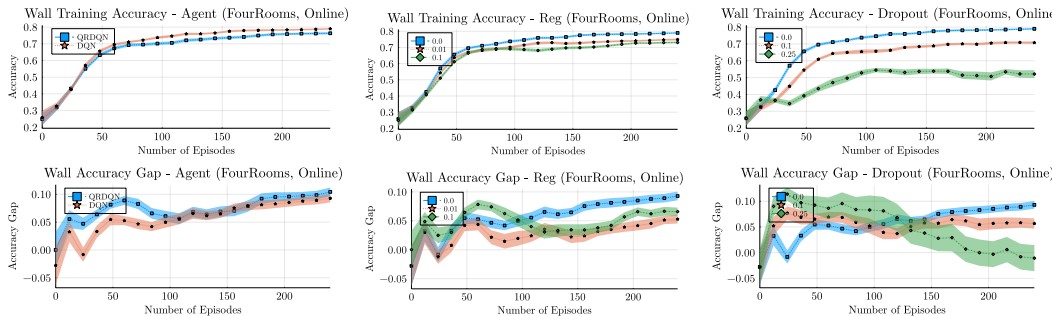

Figure 6: State generalization experiment after online learning with uncorrupted labels in the four rooms gridworld CDP. Top: Train accuracy. Bottom: Difference between testing and training accuracy on wall states. Left-Right: best agent, regularization and dropout rate respectively.

to unseen actions. We also find that, contrasting to observation generalization, dropout now improves action generalization whereas regularization does not.

### 4.3 STATE GENERALIZATION

We use the four rooms environment to evaluate an agent's ability to generalize to new states. The interior walls of the four rooms environment share either an x or y coordinate with states that the agent has encountered, and hence the student can leverage generalization between observations to these new states. We evaluate the accuracy of the policy in these interior wall states. Referring to Figure 5 and Figure 6 for offline and online training respectively, we see similar trends. In both cases, DQN is able to generalize to the new states better than QR-DQN. We also find that, despite a lower generalization gap in some setting, both regularization and dropout hurt training accuracy. This means that, although the training performance is more predictive of its testing performance, it's overall testing performance is lower.

### 4.4 DISCUSSION

Our results have validated previous findings, namely QR-DQN's superiority in the offline regime (Agarwal et al., 2020), the benefit of small amounts of L2 regularization (Farebrother et al., 2018; Igl et al., 2019; Cobbe et al., 2019). We also discover the nuanced benefit of dropout, which provides regularization in the outputs (action-values) and show that it benefits action-generalization. Our findings also show that QR-DQN's success in

observation generalization does not translate to success in state generalization, where the mean estimate of DQN is superior to quantile regression.

Our experiments require privileged information about the MDP and the ability to do dynamic programming on underlying states, and cannot be conducted on conventional environments. These insights, however, require our environment setup and ground truth quantities, but do not pose any restriction on the RL algorithm. The validation of previous findings suggest that the contextual decision process environment studied in this paper are reflective of conventional environments, such as Atari and CoinRun. The generalization challenge posed by our environments is further evidenced by the significant performance differences between training and testing (see generalization gap plots, bottom row of Figures 1-2, 4-6) across all experiments. By first demonstrating that our experimental setup confirms many previous findings, we are led to believe that our new findings would generalize to conventional environments as well.

## 5 CONCLUSION

Generalization in RL is a multi-faceted issue that deserves rigorous empirical investigation. To that end, we have presented a problem formulation using contextual decision processes and supervised learning datasets that provides ground-truth labels for optimal policies and values. Using that disentangles generalization across states, observations and actions in a single environment. By combining various gridworld environments and the MNIST dataset, we investigate generalization recover ground-truth labels for optimal policies and values through dynamic programming and investigate generalization mechanisms across states, observations and actions. We validate previous claims regarding the effectiveness of L2 regularization and QR-DQN's generalization ability in the offline setting, but find that QR-DQN cannot generalize to states as well as DQN. We also refute claims, such as dropout's ability to improve observation generalization, but find that it improves action generalization. These findings demonstrate the need for careful consideration evaluation of generalization in RL, and we hope that this line of research will continue to shed light on generalization claims in the literature. , as well as new generalization mechanisms.

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

# A APPENDIX

## A.1 EXPERIMENT SETTINGS

Code to reproduce experiments is provided in the following an anonymized dropbox link: https://www.dropbox.com/sh/ij7dvd4fr701ppe/AABf-JszLPe3AVlmaMBDsNyta?dl=0

| | Online | Online Corrupted | Offline | Offline Corrupted | Action Gen |
|---|---|---|---|---|---|
| Init Num Episodes | 50 | 50 | 50 | 50 | 50 |
| Init Policy | Random | Random | Random | Random | Custom |
| Max Episode Length | 100 | 100 | 10000 | 10000 | 10000 |
| Max Num Episodes | 300 | 300 | 100 | 100 | 100 |
| Num Online Episodes | 250 | 500 | - | - | - |
| Num Grad Steps | 1 (per env step) | 1 (per env step) | 10000 | 20000 | 20000 |
| Target Update Freq | 128 | 128 | 128 | 128 | 128 |
| Activation | ~~Relu~~ ReLU | ~~Relu~~ ReLU | ~~Relu~~ ReLU | ~~Relu~~ ReLU | ~~Relu~~ReLU |
| Hidden Size | 256 | 256 | 256 | 256 | 256 |
| Num Layers | 2 | 2 | 2 | 2 | 2 |
| Optimizer | ~~ADAM~~ Adam | ~~ADAM~~ Adam | ~~ADAM~~ Adam | ~~ADAM~~ Adam | ~~ADAM~~Adam |
| Batch Size | 128 | 128 | 128 | 128 | 128 |
| Gamma | 0.99 | 0.99 | 0.99 | 0.99 | 0.99 |
| Corruption Rate | 0 | 1.0 | 0.0 | 1.0 | 0 |

Table 1: Shared Settings for both environments, but "Action Gen" is limited to the corridor CDP. Each experiment's hyperparameters are also shared between the two models and not swept over. Custom Policy: Random except never take action $A = S - 1$ in state $S > 1$.

## A.2 ENVIRONMENT SPECIFICATION

The corridor environment has 10 states and 10 actions corresponding to the possible classification decisions the policy can make in each state. The agent begins at the right-most state of the corridor, with $s = 1$, and observes a handwritten 0 digit from the MNIST dataset. The agent must correctly classify the digit to move up to the next state. For all states, the action $a = s$ sends the agent up a state $s' = s + 1$. For a state $S > 1$, the action $a = 10 - s + 1$ sends the agent down a state $s' = s - 1$. All other actions keep the agent in the same state, but with a new observation. The reward for all transitions are $-1$ and the episode terminates when action $a = 10$ is taken in $s = 10$. The four rooms environment has four actions for each of the cardinal directions and the observation is a concatenated images corresponding to the class labels for each of the x and y coordinate. If the agent tries to move into a wall, it will stay in its current state with a new observation. Lastly, the rewards are initialized randomly (with a fixed seed) to ensure that the optimal action in each state is unique.

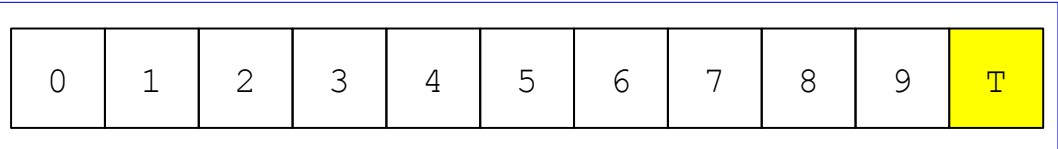

Figure 7: An illustration of the MNIST Corridor environment. Each state is labeled by a number, its index, and observations from MNIST are sampled according to this state. The initial state is state 0, and the agent is given a randomly sampled observation from MNIST corresponding to the class of handwritten zeros. There are 10 actions, one for each possible "classification" decision that must be learned through reinforcement learning. The terminal state, should by a yellow square with the letter "T", ends the episode after the correct action is taken in state 9.

Figure 8: An illustration of the MNIST FourRooms environment. Each state is labeled by two numbers. The observation given to the agent is a concatenation of a random sample from MNIST for each number in the label. For example, the initial state is 11 and so the initial observation is a concatenation of two images from MNIST corresponding to the class of handwritten ones. Although the label number is the same for the initial state, the observations can be the same or different depending on the stochastic sample. The white states are accessible to the agent but the blue states correspond to wall states which cannot be transitioned into from a white state. For state generalization, we evaluate the agent on wall states in the interior that can transition to white states: (14, 34, 41, 43, 45, 47, 54, 74).

### A.3 AGENT SPECIFICATION

Both online agents use $\epsilon$ greedy exploration with $\epsilon = 0.1$. QR-DQN computes 10 quantiles and is optimized with respect to the Huber loss, with $\kappa = 1.0$. DQN is optimized with respect to the mean squared error. The representational architecture between the agents is the same, and is shown in Table 1.

### A.4 HYPERPARAMETER SETTINGS AND EXPERIMENT PROTOCOL

The ablation ~~setings~~ settings for each experiment are detailed in the legend. ~~We~~ For every experiment, we also conducted a grid search over learning rates $\alpha = \{0.005, 0.001, 0.0005, 0.0001\}$. All experiments are averaged over 30 seeds, with a shaded region corresponding to a 95% confidence interval of the mean. The ~~best model was selected~~ results shown in each figure profile a particular generalization mechanism (DQN vs QRDQN, L2 Regularization, dropout). Each plot selects the best set of hyperparameters based on validation accuracy, which acts as a ~~proxy~~ trajectory-independent criterion for policy performance on the training environment instance. This provides an optimistic but fair assessment

of each generalization mechanism. In the corruption experiments, the best model was selected based on training accuracy.

For both the corridor and four rooms environment, we use only 10 samples from each class label to ensure that the agent can fit its training experience, but we do not limit the size of the test and validation sets. The regime where the training data is unbounded is not particularly interesting because there would be little difference between the training and testing distributions. An RL agent, in only a limited number of episodes, will only see a small fraction of the training observations, with the rest never observed (and functionally, part of the observation test set). This is unlike a supervised learning problem, where all the data is provided upfront without the need for interaction. To make the analogy to supervised learning, we were required to lower the number of training samples so that the agent would encounter a large proportion of the training observations in a limited number of episodes. This interpolative regime is also the regime of interest for the theoretical study of generalization in neural networks

We use dynamic programming to calculate the optimal policy and value function for the states, and choose to report accuracy instead of Monte-Carlo rollouts. Accuracy provides more acuity in the differences between policies, allowing us to evaluate the accuracy at each state and under any distribution. All performance criteria (training, validation and testing) are averaged over the same state distribution as the replay buffer.

## A.5    OBSERVATION GENERALIZATION IN FOUR ROOMS

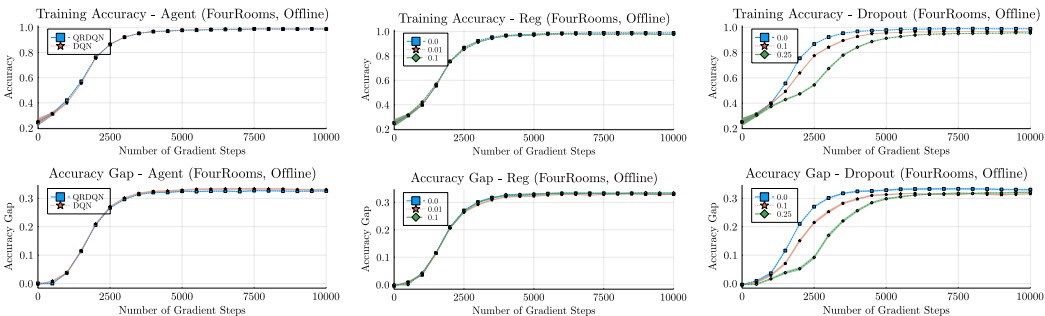

Figure 9: Observation generalization experiment after offline learning with uncorrupted labels in the fourrooms MNIST CDP. Top: Test accuracy for best agent, regularization configuration and droprate configuration. Bottom: training accuracy for each of best agent, regularization configuration and droprate.

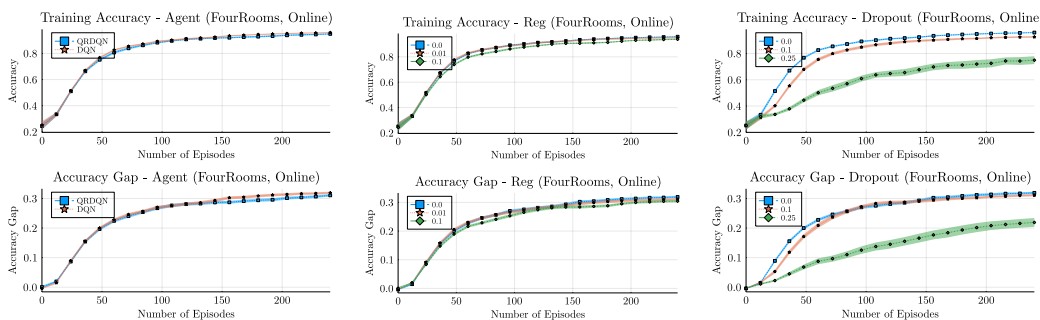

Figure 10: Observation generalization experiment after online learning with uncorrupted labels in the fourrooms MNIST CDP. Top: Test accuracy for best agent, regularization configuration and droprate configuration. Bottom: training accuracy for each of best agent, regularization configuration and droprate.

### A.6 MONTE-CARLO ROLLOUTS IN TRAINING AND TESTING ENVIRONMENTS

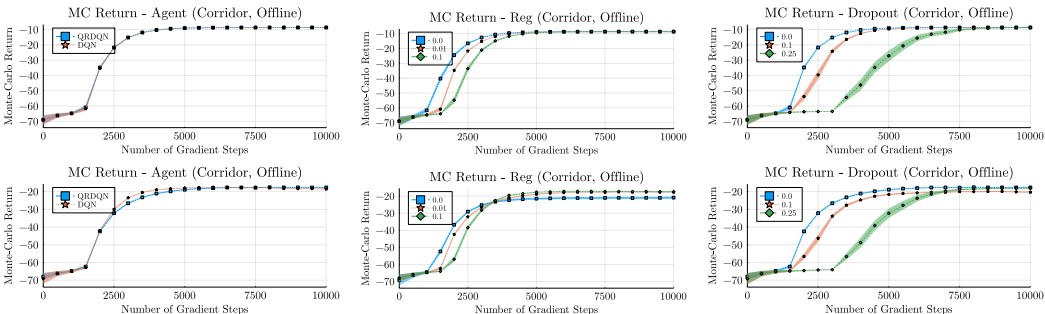

Figure 11: Observation generalization experiment after offline learning with uncorrupted labels in the corridor MNIST CDP. Top: Train MC return. Bottom: Test MC Return. Left-Right: best agent, regularization and dropout rate respectively.

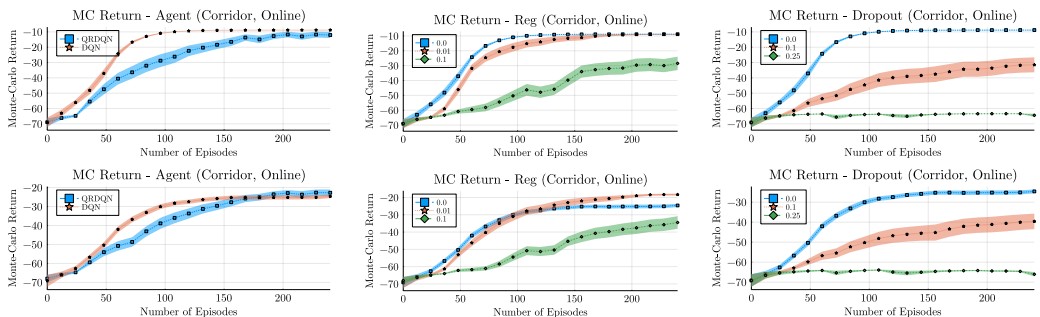

Figure 12: Observation generalization experiment after online learning with uncorrupted labels in the corridor MNIST CDP. Top: Train MC return. Bottom: Test MC Return. Left-Right: best agent, regularization and dropout rate respectively.

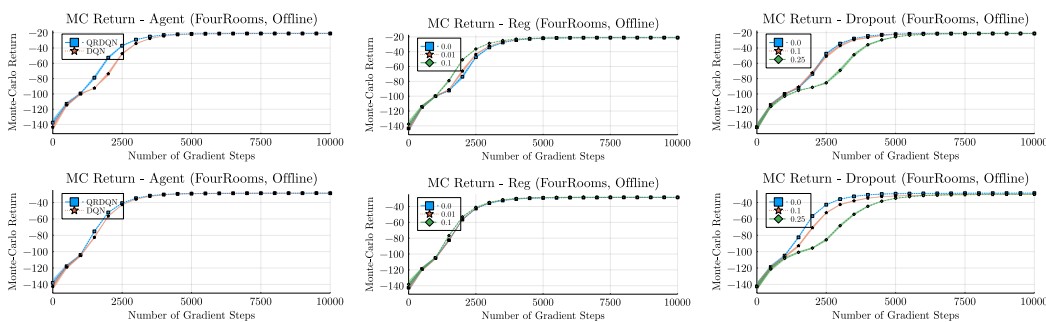

Figure 13: Observation generalization experiment after offline learning with uncorrupted labels in the fourrooms MNIST CDP. Top: Train MC return. Bottom: Test MC Return. Left-Right: best agent, regularization and dropout rate respectively.

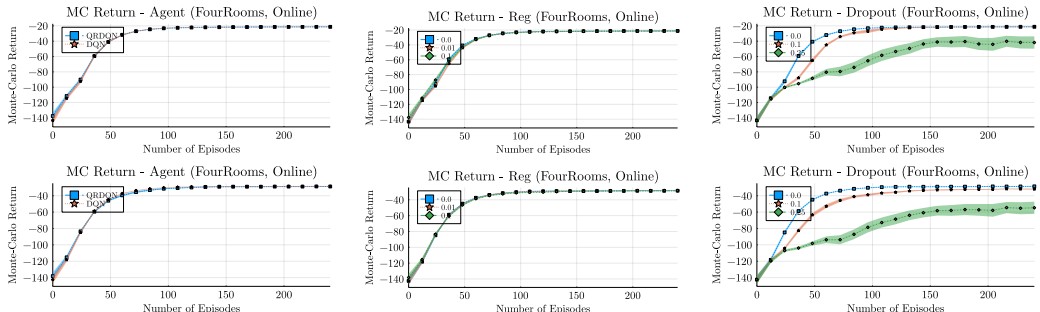

Figure 14: Observation generalization experiment after online learning with uncorrupted labels in the fourrooms MNIST CDP. Top: Train MC return. Bottom: Test MC Return. Left-Right: best agent, regularization and dropout rate respectively.

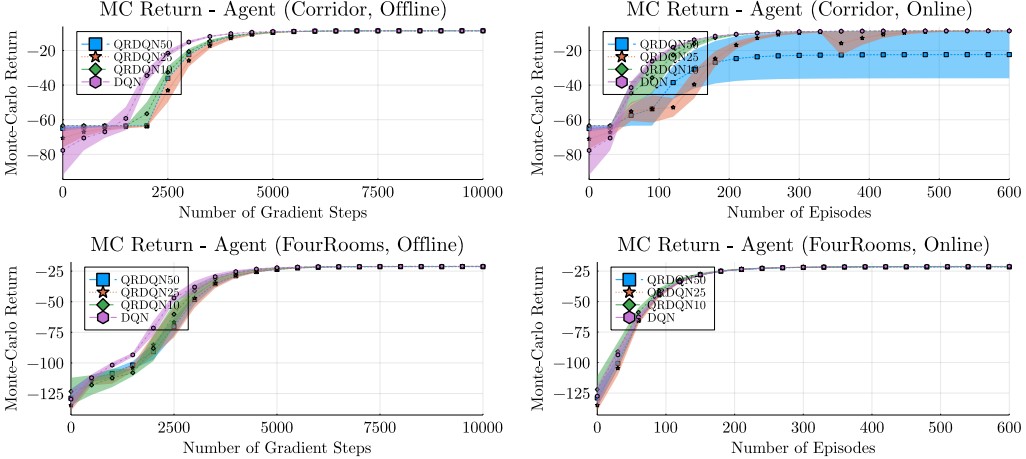

Figure 15: Corruption experiments where all MNIST labels are randomly reassigned to test whether an agent can memorize its experience. Top Left: Corridor Offline. Top right: Corridor online. Bottom Left: Four rooms offline. Bottom right: Four rooms online

