# OpenReview forum: "Disentangling Generalization in Reinforcement Learning"
_ICLR.cc/2022/Conference — ICLR 2022 Submitted_

### Official Review · Reviewer_74nY · 2021-10-31

**Correctness:** 3
**Technical Novelty And Significance:** 3
**Empirical Novelty And Significance:** 2
**Recommendation:** 5
**Confidence:** 4

**Main Review:**

This paper has multiple strong points. Authors propose a new perspective to evaluate generalisation in RL agents based not on accumulated rewards as most research usually do now, but through a careful evaluation of the generalisation performance of the different elements of an MDP. From the theoretical point of view, most of the mathematical constructions are drawn from previous works (well referred). Still, this work presents a new empirical formulation to better study generalisation.
My main concern is that this paper is not the first pointing to change in this manner the way we evaluate generalisation and, the empirical method prosed here is very limited, in the sense that it requires full knowledge of the state space and transition dynamics, e.g., it uses dynamic programming to recover optimal value and policy. Additionally, the empirical approach used in this work, mapping the states to observations of the MNIST dataset, illustrates that this whole evaluation is carried on relatively small hand-crafted environment. It leaves me doubtful if the results obtained here, e.g., that dropout only helps action generalisation, would really translate to more visually rich and complex domains. Note that authors also state that the RL algorithms were memorising the training data, I wonder what would happen if this was not the case.

Minor mistake/suggestions:
* End of page 2: “The MNIST gridworld environment used by Lee et al. (2018) may be considered a CDP, but is not recognized as such” why?
* There where some points of the literature review that became obvious only after reading the whole work, it might be beneficial moving this section to the end
* Section 2, authors don’t present the bellman equations in standard form but as expectations, it would be beneficial for the reader to anticipate here why this is relevant in this work.
*Section 2.1 second paragraph, it is not clear to me why rewards cannot be a good way to measure generalisation in the environment, (if you don’t know if your policy is optimal, only way to reduce uncertainty when you don’t have full access to the MDP dynamics is to keep checking new policies to whether one of them produces better reward)
*Section 4, a figure of the environments showing how they are used with the MNIST dataset, would help here
* Section 4.1, line 4. Figure 10 should be Figure 1
* Figure 1, it is not specified which of the two RL algorithms is used while evaluating the regularisation procedures.


**Summary Of The Paper:**

This paper proposes a new methodology to test the generalisation performance in reinforcement learning. Authors study two  (non state-of-the-art) methods (DQN and QR-DQN) and analyse how L2 regularisation and dropout affect the ability of the algorithms generalising to new observations, states and actions (each of them independently)

**Summary Of The Review:**

Authors explore a non-standard method to measure generalisation in RL applying it empirically to two deep RL algorithms. Despite of authors multiple points against using rewards as generalisation metric, I am still not convinced that the struggles of RL with respect to supervised learning, which is the absence of the ground truth answer, is tackled by the proposed method. Authors rely on heavy assumptions of the environment, allowing them to have access to the "ground truth", i.e., optimal policies and value. This is not the case in most problems about RL generalisation.

---

> ### Author Response · Authors · 2021-11-21
> **Response Pt 1**
>
> Thank you for your review! Below we address the points that you raised:
>
> "Minor" mistakes and suggestions
> --------------------------------
>
> The list of minor mistakes and suggestions had some very interesting
> points. We address these in order, but draw particular attention to the
> third and fourth bullet point:
>
> -   Lee et. al (2018) proposed a simple gridworld environment that uses
>     MNIST images as a state representation to minimize correlation
>     between neighboring states. There is a fixed deterministic map
>     between a particular state and a particular pair of MNIST images,
>     which makes it a simple contextual decision process. The
>     observations are not stochastically sampled however, and there is no
>     distinction between observations and state. The paper is also not
>     concerned with generalization, and so do not use ground truth
>     quantities or partition between train/test.
> -   We think that having the literature in the introduction is important
>     to contextualize our contribution. We will make some changes to make
>     the points more clear without the later sections.
> -   **(Bellman Equation)** Equation 1 is not a bellman equation, but the
>     definition of value and action-value. These are not the usual
>     definition in the literature, which are expectations over a
>     trajectory. We use the definition that takes the expectations over
>     the stationary state distribution. This is important because the way
>     in which we measure the many facets of generalization in our
>     experiments is through an expectation over the state distribution
>     (e.g. optimal action accuracy, MSE), and not the trajectory.
> -   **(RL Rewards/Return)** Without any knowledge about the MDP or
>     environment, the sum of rewards is the only criterion possible to
>     measure the quality of a learned policy. As an experimenter however,
>     it is limiting: it is trajectory dependent (which conflates
>     generalization across states/actions/observations) and highly
>     stochastic (either from the policy or the environment, or both).
>     Using our environment and experiment setup, we are able to
>     disentangle generalization by introducing a more general set of
>     criteria that are also not trajectory dependent. As we show,
>     different generalization mechanisms improve generalization across
>     different axes. We are also able to get strong statistically
>     significant results with only 30 runs because we do not rely on the
>     stochasticity of a Monte-Carlo rollout.
> -   We will include a figure in the appendix illustrating the
>     environments.
> -   This is correct. Figures 10 and 14 in the appendix had the same
>     label as the one in the main body, and this unfortunately created
>     incorrect Figure references in the main text. We have fixed this.
> -   Each figure summarizes an experiment with a grid search over all
>     possible combinations of algorithm, dropout rate, L2 regularization
>     and learning rate. The results presented, e.g. for dropout,
>     correspond to the best possible combination of all other components.
>     This provides an optimistic but fair assessment. We will include
>     figures in the appendix that control for the other components, but
>     remark that this generates 18 figures (each with 6 plots) per
>     experiment (which we have 6 experiments in total). We will also
>     update the Dropbox code with instructions on how to do this.
>
> "… Not the first pointing to change in this manner the way we evaluate generalization"
> --------------------------------------------------------------------------------------
>
> As we discuss in the introduction, there is a growing body of research
> that either: proposes an algorithm to improve "generalization" or
> proposes environments that can be partitioned into testing and training
> sets to measure "generalization". We argue that both of these
> approaches, while valid, are more akin to transfer learning than
> generalization. Our approach is uniquely analogous to generalization in
> supervised learning, and shows the importance of this analogy by
> disentangling generalization across states, observations and actions.
> The "experimental lens" that our paper provides, and the results that we
> are able to obtain, are thus significantly different from previous work.

---

> > ### Comment · Reviewer_74nY · 2021-11-25
> > **Thank you for your response**
> >
> > I want to thank authors for their detailed response and clarifications. I think the clarifications strength the submission.
> >
> > I would like to clarify that when I mentioned that authors are not the first pointing to change how we measure generalisation I meant that the critic to the current approach and proposal to move in this direction is not new, although, as I stated in the original review, the specific  manner that authors propose and the experimental analysis is novel. I also think the problem attempted is important and challenging. That said, my main concerns remain as I detail below and that's why I am keeping my score in a 5.
> >
> > I agree with authors that environments like Atari aren't necessarily more meaningful (if anything ProcGen would be), and multiple interesting new ideas have recently been presented in MiniGrid for instance, which uses procedural grid worlds.
> >
> > My concern is that while those environments also use finite number of states, they provide harder benchmarks which the agents are not able to "memorise". Note also that, to avoid that this "memorisation" happens we cannot rely on a finite number of maps. Instead, environments need to be procedurally generated and there is a plethora of works now on how to generate such environments without just "random" designing to boost learning. I don't see trivial to apply the form of analysis authors propose here to those complex problems, and restricted to simple memorisable scenarios the proposed method does not really tell how broad the findings here are.  I believe that if authors were able to make their analysis to work in complex procedural environments like the ones generated in [1] (which are also using finite grid worlds) it would dramatically change the view of reviewers on this.
> >
> >
> > Without that, authors are presenting broad claims e.g. "dropout does not help with observation generalisation" with a single and very constrained environment. This dramatically reduces the confidence of the reader.  Moreover, the experiment requires of the authors to design a mapping function of each environment's observation to an image from MNIST, thus the agent's is handling a "proxy" of the real problem to solve, and I am a concerned on how this is affecting the learning process of the agent. As we know from learning from rewards, any proxy that we add can make the agent diverge completely from our original objective. Thus, I wonder if the effect of dropout would be the same, with the agent learning with the real observations from the environment, instead of the MNIST proxy.
> >
> >
> > [1] Campero, Andres, et al. "Learning with AMIGo: Adversarially Motivated Intrinsic Goals." International Conference on Learning Representations. 2020.

---

> > > ### Author Response · Authors · 2021-11-29
> > > **Appreciate your clarifications, but do not agree with the fundamental argument**
> > >
> > > We appreciate your clarifications, but do not agree with the fundamental
> > >     argument underlying this critique. The direction that we explore,
> > >     generalization in a single environment, does not currently exist in the
> > >     literature. This evidenced by the striking lack of work exploring a natural
> > >     characterization of generalization RL: between states, observations and
> > >     actions. The reason for this is that current environments, including
> > >     procedurally generated environments, do not allow us to separately measure
> > >     an agent's ability to generalize to new states, observations and actions.
> > >     Instead, generalization in RL is primarily concerned with the expected sum
> > >     of reward in a separately generated environment. The point made about
> > >     procedural generated environments is what this work is explicitly avoiding.
> > >
> > > The term memorization in our context is different from the context that you
> > >     are using. Minigrid, and other procedurally generated environments, only
> > >     provide states; they do not serve observations. Memorizing states is
> > >     problematic, but this is not what our agent is doing. The agent in our
> > >     environment can fit the training observations it has seen. But these
> > >     observations are not seen again during testing, because during testing the
> > >     observations come from the test set. This allows us to separately evaluate
> > >     generalization across observations without varying the underlying state or
> > >     the dynamics. This feature is pivotal to our experiments and the
> > >     characterization of RL that we propose. This cannot be done in Minigrid
> > >     without also using the observation structure that we use in our paper.
> > >
> > > There is no proxy problem that we are solving. The MDP that we care about is
> > >     exactly the MDP that produces observations from the MNIST dataset through an
> > >     emission function. There is no other observation being produced by the MDP,
> > >     and learning in the underlying tabular MDP is irrelevant.

---

> ### Author Response · Authors · 2021-11-21
> **Response Pt 2**
>
>
> Empirical method is limited and generality concerns, especially for visually rich environments
> ----------------------------------------------------------------------------------------------
>
> To be specific, we are limiting our study to environments with finite
> state space. There is no restriction on the dataset, and we only enforce
> the disjoint property on emission function that maps states to
> observations. This restriction is necessary to conduct our empirical
> study, because knowledge of ground-truth values is not available in
> other environments. While you expressed some skepticism at using
> criteria other than the sum of reward, we would like to state again that
> the sum of reward is trajectory dependent and conflates generalization
> across states, observations and actions.
>
> We would argue that the environments are not "hand crafted" in the sense
> that they were designed for the task at hand. They are simple however:
> we use common MDPs in the literature (short corridor and four rooms) and
> a common dataset (MNIST). This is by design. Although both the tabular
> MDPs and MNIST are independently well-studied, many interesting
> phenomena can be studied at their intersection, as this study
> demonstrates.
>
> While the underlying MDP in our experiments is limited to a finite state
> space, this is not an unrealistic assumption. Games, including Atari
> with only 128 bytes of ram, have large but finite state spaces.
> State-generalization is already challenging in the FourRooms MDP (as
> shown by the suboptimal training accuracy in Figure 6, top), and this
> difficulty will only increase with larger state-spaces.
>
> Moreover, the assumptions in our experimental setting are necessary to
> balance simplicity and complexity. We need simplicity in the states to
> recover groundtruth quantities through dynamic programming. Yet, we need
> complexity in the observations for agents to learn generalizable
> representations. The construction presented in this paper achieves this
> balance, as evidenced by the statistically significant differences
> between train and test, and the findings corroborating (L2, QRDQN) and
> refuting (dropout) previous claims of generalization in observation
> space.
>
> Generalization without RL agent's memorizing the training data
> --------------------------------------------------------------
>
> Earlier experiments were conducted where the number of training
> observations was not limited. This regime is not particularly
> interesting because there is little difference between the training and
> testing distributions. An RL agent, in only a limited number of
> episodes, will only see a small fraction of the training observations,
> with the rest never observed (and functionally, part of the observation
> test set). This is unlike a supervised learning problem, where all the
> data is provided upfront without the need for interaction. To make the
> analogy to supervised learning, we were required to lower the number of
> training samples so that the agent would encounter a large proportion of
> the training observations in a limited number of episodes. This
> interpolative regime is also the regime of interest for the theoretical
> study of generalization in neural networks (Neyshabur et al. 2017).
>
> References
> ----------
>
> Neyshabur, Behnam, Srinadh Bhojanapalli, David Mcallester, and Nati
> Srebro. 2017. “Exploring Generalization in Deep Learning.” In *Advances
> in Neural Information Processing Systems*, edited by I. Guyon, U. V.
> Luxburg, S. Bengio, H. Wallach, R. Fergus, S. Vishwanathan, and R.
> Garnett. Vol. 30. Curran Associates, Inc.
> <https://proceedings.neurips.cc/paper/2017/file/10ce03a1ed01077e3e289f3e53c72813-Paper.pdf>.

---

### Official Review · Reviewer_iELT · 2021-11-01

**Correctness:** 3
**Technical Novelty And Significance:** 2
**Empirical Novelty And Significance:** 2
**Recommendation:** 5
**Confidence:** 4

**Main Review:**

Strength:

- The paper discussed an important problem in RL. The writting is clear and easy to follow.

- The exprimental results are abundunt and there are enough repeats (30) to ensure statistical significance. The plots are well presented.

Major Concerns:

- The largest concern is that the experiments are limited to simple grid world envrionments with an unnatural setting to use MNIST images as observation. In particular, the authors wrote in Chap 2.2 "If the total number of classes K is equal to the number of states |S| in the MDP, then each state can be uniquely identified with a class label". This assumption restricts the scope of this study to discrete and relatively small state space. Since the results are mostly empirical, it remains unclear how these results generalize to more practical and real-world RL tasks.

- The novelty and significance of technical conritbution is limited. Either the concept of CDP or using performance gap between train and test set as evaluation criteria is not original. The being studied L2 regularization and dropout have also been discussed in Cobbe et al, 2018 (actually should be 2019 as the publication year of ICML).

- Although the experimental results are clear and easy to understand, I expect a more comprehensive empirical investigation, e.g.,  visualizing the learned policy in your envrionments (as in Cobbe et al, 2018), and perform experiments in more tasks.


Other issues:

- In the end of Chap 4.1, "Referring to 14,", what is 14? Do the authors mean Figure 14?
- In "Figure" 7 (should be a table): Relu -> ReLU, ADAM -> Adam.
- The spatial interval between Figure 5 and 6 is too narrow.
- In chap 4, before 4.1, it was written "some have reported that these techniques improve generalization.". Are there some references?
- References are not professional, please check the venue of publications instead of just citing the arXiv version.



**Summary Of The Paper:**

This paper discusses generalization in deep RL. The key contribution of the paper, from my understanding, is that the authors argue that different from generalization in SL, in RL state, observation and action should be considered separately. A measurement (Eq.4) is proposed to evaluate generalization capacity of deep RL within the contectual decision process (CDP) scheme. Experiments were performed on grid world environments with MNIST image as observations and several results were concluded.


**Summary Of The Review:**

Overall, the paper discussed an significant topic in deep RL and their paradigm for evaluation is clearly presented and empirically investigated. However, the current study restricts itself to simple grid world envrionments with an unnatural setting to use MNIST images as observation. While the main contribution of this paper is empirical, it remains unclear how these experimental results generalize to more practical and real-world RL tasks. Also, the technical novelty and significance is limited. In sum, I consider the current paper fail to meet the acceptance criteria of ICLR and recommend a major revision, probably by performaning more comprehensice investigation on more tasks with preferrably continuous state space.


--------- Post Rebuttal -----------

The authors have addressed most of my concerns, and I increased my score by one level accordingly. However, a core limitation of the current work, "do the results/insights also apply to more realistic environments with continuous/large state space?", is not resolved. While I believe this work has its potential, currently I lean toward weak reject.

---

> ### Author Response · Authors · 2021-11-17
> **Response**
>
> Thank you for your review! Below we address the points that you raised:
>
> Limited to grid world and MNIST
> -------------------------------
>
> To be specific, we are limiting our study to environments with finite
> state space. There is no restriction on the dataset, and we only enforce
> the disjoint property on emission function that maps states to
> observations. The quote
>
> "If the total number of classes K is equal to the number of states \|S\|
> in the MDP, then each state can be uniquely identified with a class
> label"
>
> is an example of a possible instantiation of a contextual decision
> process given a dataset. In addition, even this example contextual
> decision process is not restrictive. Given any MDP with \|S\| states, we
> can pool the observations from numerous datasets, such that the total
> number of classes across all datasets can be mapped to the MDP states.
> Note, we also give another example in 2.2, where the states in a
> gridworld, given (x,y) coordinates, are mapped for each x and y
> component separately. There are many possibilities, but these are two
> examples that we choose tn study because of their analogy to
> well-studied RL problems.
>
> While the underlying MDP in our experiments is limited to a finite state
> space, this is not an unrealistic assumption. Games, including Atari
> with only 128 bytes of ram, have large but finite state spaces.
> State-generalization is already challenging in the FourRooms MDP (as shown
> by the suboptimal training accuracy in Figure 6, top), and this
> difficulty will only increase with larger state-spaces.
>
> Moreover, the assumptions in our experimental setting are necessary to
> balance simplicity and complexity. We need simplicity in the states to
> recover groundtruth quantities through dynamic programming. Yet, we need
> complexity in the observations for agents to learn generalizable
> representations. The construction presented in this paper achieves this
> balance, as evidenced by the statistically significant differences
> between train and test, and the findings corroborating (L2, QRDQN) and
> refuting (dropout) previous claims of generalization in observation
> space.
>
> CDP, L2 Regularization, dropout previously studied
> --------------------------------------------------
>
> While the contextual decision process is a previously established
> theoretical setting, it has not been used to study generalization as in
> our paper. The different axes of generalization in RL have also not been
> characterized.
>
> In regards to the generalization mechanisms studied in the empirical
> section, our work carefully does not introduce new mechanisms. Instead,
> we aim to provide an "experimental lens" to corroborate or refute
> current findings in the literature. This experimental lens provides a detailed view of generalization, which is not provided by
> generalization studies in conventional RL environments. The environments
> studied in this paper constitute a minimal setting in which questions of
> generalization can be precisely probed. We believe that further new
> mechanisms can be teased apart from our proposed disentangling of
> generalization in RL. Before that can be done however, the merit of our
> proposed "experimental lens" must be evaluated using current claims of
> generalization in the RL literature.
>
> More comprehensive empirical study, visualizing learned policy
> --------------------------------------------------------------
>
> The empirical study is comprehensive in its study of generalization, and
> it is unclear what type of problem is left unaddressed. We cover
> generalization across three different axes, two of which (action and
> state) have not been studied in detail or with comparison to other axes
> of generalization. The two environments accentuate the differences in
> action and state generalization. Moreover, the results already show
> statistically significant differences between the three potential
> generalization mechanisms (L2, Dropout, QRDQN).
>
> It is also unclear what visualizing the learned policy will show that is
> not shown in the summary plots. We cannot simply overlay the action chosen by the policy in each state because the actions are determined by the observation, which are sampled at each state. However, our experimental results do
> include results for state-specific quantities (code link in appendix). For
> example, we have accuracy and MSE for each state individually, averaged over observations. We can
> obtain summary statistics using any distribution over states (i.e.
> uniform, optimal policy, epsilon-optimal, training distribution, etc.).
> We did not include the individual state figures in the appendix because,
> even just for the 10 state corridor, this corresponds to over 40
> figures.
>
> Other issues
> ------------
>
> Thank you for bringing these to our attention. The incorrect figure
> labels and typos have been fixed, and additional references have been
> included to qualify statements. We have also updated the references for
> those that have been published at a peer-reviewed venue.

---

> > ### Comment · Reviewer_iELT · 2021-11-22
> > **Thanks for the authors' reply**
> >
> > The rebuttal have tried to address 3 of my major concerns. While the author has resolved my last concern, I am not convinced for the first 2 concerns (scalability and novelty).
> >
> >
> > > While the underlying MDP in our experiments is limited to a finite state space, this is not an unrealistic assumption. Games, including Atari with only 128 bytes of ram, have large but finite state spaces.
> >
> > Although theoretically Atari has finite state spaces, but $2^{128}$ is too large and I feel the current method will be difficult to tackle with such large space.
> >
> > > Moreover, the assumptions in our experimental setting are necessary to balance simplicity and complexity. We need simplicity in the states to recover groundtruth quantities through dynamic programming.
> >
> > As the current proposed method need simplicity in the states, I believe that some new methods need to be developed to improve scalability of the current work
> >
> > > In regards to the generalization mechanisms studied in the empirical section, our work carefully does not introduce new mechanisms. Instead, we aim to provide an "experimental lens" to corroborate or refute current findings in the literature.
> >
> > While there are empirical contributions (though need state space to be simple as the authors mentioned), I still consider the novelty and significance of technical conritbution of the current paper is limited, since no new mechanisms is introduced.

---

> > > ### Author Response · Authors · 2021-11-22
> > > **We are not proposing an algorithm for use in Atari, scalability concerns are not applicable to this work.**
> > >
> > > We are not proposing an algorithm for use in Atari, scalability concerns are not applicable to this work. We feel that the contribution of this work is still being mischaracterized. In response, we would like to reiterate the following two points:
> > >
> > > - We are not proposing a new algorithm. Succinctly, we 1) provide a new characterization of generalization in reinforcement learning, 2) provide a new empirical study showing the importance of this characterization.
> > >
> > > - Our experiments investigating our characterization of generalization in RL shows that different generalization mechanisms benefit different generalization axes. *These empirical observations could not be obtained in conventional environments, like Atari.* While our experiments are limited to a (broad) class of finite state/action MDPs, our insights could not be obtained without this restriction.
> > >
> > > In summary, we are not proposing a new algorithm that is intended to be used in Atari. We are proposing a new characterization of generalization in RL and the first empirical study that shows the important differences between generalization across states, observations and actions. These experiments cannot be run in Atari because they rely on priveleged information about the MDP. The experimental results nonetheless shed insight on the RL algorithms that we are using in environments like Atari. This is a unique perspective, because there is no restriction on the RL algorithms. We merely restrict the environments to allow us, as researchers, to better probe the generalization abilities of these RL algorithms.

---

> > > > ### Comment · Reviewer_iELT · 2021-11-22
> > > > **Clarification**
> > > >
> > > > > We are not proposing a new algorithm.
> > > >
> > > > I understand that the authors are not proposing a new algorithm for solving a task (I did not use the words "algorithm" in my comments).  As I wrote in the summary, "The key contribution of the paper, from my understanding, is that the authors argue that different from generalization in SL, in RL state, observation and action should be considered separately." My concern of "scalability" is that **do the results/insights also apply to more realistic environments with continuous/large state space?**  As current methodology of evaluating generalization needs the state space to be simple, this cannot be answered.
> > > >
> > > >
> > > > > we are not proposing a new algorithm that is intended to be used in Atari.
> > > >
> > > > I was not asking the authors to do Atari experiments. In my original review, I did not asked to use Atari (I said " more practical and real-world RL tasks.", and I did not think Atari is real-world RL task). Instead, in the rebuttal of the authors, it was mentioned "While the underlying MDP in our experiments is limited to a finite state space, this is not an unrealistic assumption. Games, including Atari with only 128 bytes of ram, have large but finite state spaces".  This sounds like the authors are claiming their evaluation methods can apply to Atari.
> > > >
> > > > Overall, I like the work for the conceptual novelty and rigorious empirical studies. Meanwhile, I  am not convincible that the current paper meets the acceptance criterias because there are only empirical insights in grid worlds, and the technical contribution is limited since the paper does not introduce new mechanisms for studying generalization.
> > > >
> > > > I believe that there are large spaces to improve the significance of the current work by proposing new analysis mechanims. But before that, in consistence with other reviewers, I lean toward rejection.

---

> > > > > ### Author Response · Authors · 2021-11-22
> > > > > **Thank you for the clarification!**
> > > > >
> > > > > Thank you for the clarification! First, we apologize for the confusion: our comment on Atari was meant to signify that Atari exists within the class of environments that we are investigating. We can expect that our results could generalize better within this family of environments (finite state/action MDPs), rather than a dynamical system with continuous state/aciton.
> > > > >
> > > > > As for the heart of your concern: it is of course difficult to say whether our results would hold in any specific "more practical and real-world RL task," given that our experiment setup cannot be replicated in these convential environments. This submission does, however, show that many currently held beliefs about existing generalization mechanisms and RL algorithms also hold in our proposed experimental study. This common-ground is used to suggest that the environments that we study are indeed similar to the more complex environments studied elsewhere. We use this as a launching ground to show the intricate effects between generalization mechanisms (including choice of RL algorithm) and the different axes of generalization. By first demonstrating that our experimental setup confirms many previous findings, we are led to believe that our new findings would generalize as well.
> > > > >
> > > > > We acknowledge that this argument is not explicit in our submission, and we will be uploading a revision tomorrow with the suggested changes from you and the other reviewers.

---

### Official Review · Reviewer_1S93 · 2021-11-03

**Correctness:** 3
**Technical Novelty And Significance:** 2
**Empirical Novelty And Significance:** 2
**Recommendation:** 3
**Confidence:** 4

**Main Review:**

I like the theme of the paper, and I indeed agree that designing the right metrics for quantifying generalization in single-task RL are needed and are not quite present in the literature. However, I think the paper falls short of delivering on and convincing the reader that the empirical analysis is useful and generalizable, and the proposed metric is a good one. I will explain why:

1. It is unclear how these observations transfer to other RL problems. Would testing generalization (within a single task) on other domains, e.g., Atari get the same results? Why or why not? For example, if the claim is that dropout doesn't help, does it not help on this task, or does the analysis also support why it shouldn't help on other domains?

2. One good point about converting SL problems to RL problems is that it is easy to measure generalization against a groundtruth SL oracle. With that in mind, the performance of RL agents is worse, so one argument is that everything is generalizing poorly on an absolute scale. Comparing with such metrics should have been more informative, compared to just DQN and QR-DQN, and this is a limitation.

3. In the offline setting, it is known that better algorithms that are more uncertainty-aware and do not commit to a single target value, such as ensemble DQN, random ensemble mixture, QR-DQN, etc work much better compared to the DQN, because they can be more accurate on OOD predictions. Does the action generalization experiment say essentially this? What is the new insight behind this high-level known result?

4. What insights should I take away as to why one algorithm is worse than the other in each of the different settings? Does it indicate that some principle is behind the relative performance differences? A detailed empirical analysis of what about QR-DQN and what about DQN makes them perform differently in these different generalization challenges would shed light on this. Similarly, a rigorous empirical study of why dropout helps or why L2 helps, etc, seems necessary to me to take away from this paper.

5. How does the accuracy translate to metrics I can measure during training, such as some kind of loss, validation error, etc? This is essential in understanding how one should modify algorithms to make them generalize better.



**Summary Of The Paper:**

This paper proposes an approach to measure to quantify generalization properties (state generalization, observation generalization, action generalization) in single-task RL in the context of offline RL. The paper discusses the limitations of several existing approaches for measuring cross-environment generalization, then presents their generic measure of generalization (Equation 4), and evaluates generalization when learning from offline data using a DQN and QR-DQN in a contextual decision process (CDP) problem created out of MNIST classification. The results suggest that dropout is effective for action generalization, not state generalization, and L2 penalty is effective, and QR-DQN can generalize better than DQN in the offline setting, but worse in terms state generalization.

**Summary Of The Review:**

Without presenting any analysis of the **why** question, it seems that the work is not complete in my opinion. Also, the generalizability of these findings to other domains is under question. If these two major concerns can be addressed, especially along the lines of the pointers above, I am happy to revise my score. But for now, I would vote for rejection of the paper.

---

> ### Author Response · Authors · 2021-11-15
> **Response Pt 1**
>
> Thank you for your review! Below we address the points that you raised:
>
> Unclear how the empirical findings generalize
> ---------------------------------------------
>
> Our work carefully does not introduce new mechanisms, but provides an
> "experimental lens" to corroborate or refute current findings in the
> literature. The claims that we test, and our subsequent empirical
> findings, are general in the sense that neither the environments nor
> mechanisms (distributional RL, dropout, L2) are tailored to any specific
> RL challenge (e.g. "hard" exploration). The environments studied in this
> paper constitute a minimal setting in which interesting questions of
> generalization can be probed.
>
> It is impossible to conduct the same empirical study in Atari, or any
> other RL environment, because of the lack of ground truth RL quantities
> and the inability to partition states, actions and observations. While
> one might hope that certain Atari games highlight the benefit of a
> particular axis of generalization, it is difficult to make that
> assessment without the disentangled evaluation that our study provides.
> Given that it is impossible to conduct the same empirical study in
> Atari, we remark that Atari itself is a contextual decision process with
> state being RAM and with pixel observations being emitted. Of course,
> the contextual decision process of Atari is more complicated in both the
> underlying MDP and the observations.
>
> We believe that these findings are general and that further new
> mechanisms can be teased apart from our proposed disentangling of
> generalization in RL. Before that can be done however, we believe that
> the merit of our proposed "experimental lens" must be evaluated using
> current claims of generalization in the RL literature.
>
> Comments on generalization with groundtruth SL oracle
> -----------------------------------------------------
>
> "One good point about converting SL problems to RL problems is that it
> is easy to measure generalization against a groundtruth SL oracle."
>
> To clarify this comment, we do not convert an SL problem to an RL
> problem. We are using a classification dataset for an RL problem, which
> is specified by the MDP. We also do not use a supervised learning oracle
> but a reinforcement learning oracle. The ground truth quantities, such
> as the optimal action (for accuracy) or true action-value (for MSE), are
> obtained using dynamic programming on the underlying MDP.
>
> "Comparing with such metrics should have been more informative, compared
> to just DQN and QR-DQN, and this is a limitation."
>
> It is not entirely clear what the reviewer means by this statement.
> Supervised learning is impossible in this empirical study, because the
> problem is still a reinforcement learning problem (i.e. to maximize the
> return).
>
> Offline setting and methods that quantify uncertainty
> -----------------------------------------------------
>
> We are aware that uncertainty quantification and probabilistic methods
> have been used to the benefit of offline RL. This is not conclusive
> however, and there are signs that effective methods can be much simpler
> (Fujimoto and Gu 2021). Because of Dropout's connection to variational
> inference (Gal and Ghahramani 2016), our findings does lend credence to
> the effect of uncertainty quantification as a regularizer for action
> uncertainty. This is an interesting connection, and we will highlight
> this fact in the paper.

---

> > ### Author Response · Authors · 2021-11-15
> > **Response Pt 2**
> >
> > Overall intuitions for results
> > ------------------------------
> >
> > As discussed in the shared reply, we did not think it correct to include
> > intuition on the nature of our results without room for further
> > experimentation or connection to already established literature. Below,
> > we speculate on our findings.
> >
> > Between L2 regularization and dropout, we find the surprising fact that
> > dropout provides action generalization benefits and L2 regularization
> > helps observation generalization. This can be explained by the fact that
> > L2 penalizes weights globally across all inputs, which favors smoother
> > solutions between observations/states. This explains the improvement of
> > observation generalization across all experiments. Even though the
> > training performance on unseen states (Figure 6, top middle) is lower
> > with L2 regularization, the smoothness of the solution still improves
> > the generalization gap (Figure 6, bottom middle). Dropout, on the other
> > hand, provides local regularization for each input by turning off random
> > weights. To compensate, the outputs in action space must be smoother but
> > this smoothness does not occur between inputs.
> >
> > Between QR-DQN and DQN, we see in Figure 3 that QR-DQN can struggle to
> > fit the corrupted training data as the number of heads increases. This
> > means that, although QR-DQN has more capacity because of the number of
> > heads, this capacity must be spent to predict more quantiles of the
> > return distribution. The heads provide an inductive bias that helps
> > generalization when data is limited, as in the offline regime (Figure 1,
> > left). These heads can overfit however, as evidenced by the relatively
> > worse action-generalization (Figure 4, left). This suggests that, in the
> > online regime (Figure 2, left), QR-DQN is not able to use the
> > distributional inductive bias to improve generalization because the
> > heads are constantly changing their estimates of the training
> > distribution for return as new transitions are encountered.
> >
> > Translating metrics to training heuristics
> > ------------------------------------------
> >
> > Including training heuristics for other environments is an important
> > next step. It is not possible to do this exactly, as having ground truth
> > values in any environment supplants the need for RL in that environment.
> > Current work on auxiliary tasks may help provide pseudo groundtruth
> > values (Lyle et al. 2021, Zheng et al. 2021) to proxy for
> > generalization, but this is far outside the scope of this paper.
> >
> > "Answering the why question"
> > ----------------------------
> >
> > We feel that the "why" question is answered in our submission: the current experimental approach does not disentangle the many ways an RL agent can generalize and this is equally reflected in the definitions of generalization studied in the RL literature. We hope
> > that this rebuttal clarifies that our approach fills this gap, but please let us know if you
> > feel that we mischaracterized or missed one of your concerns.
> >
> > References
> > ----------
> >
> > Fujimoto, Scott, and Shixiang Shane Gu. 2021. “A Minimalist Approach to
> > Offline Reinforcement Learning.” *arXiv:2106.06860*.
> > <http://arxiv.org/abs/2106.06860v1>.
> >
> > Gal, Yarin, and Zoubin Ghahramani. 2016. “Dropout as a Bayesian
> > Approximation: Representing Model Uncertainty in Deep Learning.” In
> > *Proceedings of the 33nd International Conference on Machine Learning,
> > ICML 2016, New York City, Ny, Usa, June 19-24, 2016*, edited by
> > Maria-Florina Balcan and Kilian Q. Weinberger, 48:1050–9. JMLR Workshop
> > and Conference Proceedings. JMLR.org.
> > <http://proceedings.mlr.press/v48/gal16.html>.
> >
> > Lyle, Clare, Mark Rowland, Georg Ostrovski, and Will Dabney. 2021. “On
> > the Effect of Auxiliary Tasks on Representation Dynamics.” In *The 24th
> > International Conference on Artificial Intelligence and Statistics,
> > AISTATS 2021, April 13-15, 2021, Virtual Event*, edited by Arindam
> > Banerjee and Kenji Fukumizu, 130:1–9. Proceedings of Machine Learning
> > Research. PMLR. <http://proceedings.mlr.press/v130/lyle21a.html>.
> >
> > Zheng, Zeyu, Vivek Veeriah, Risto Vuorio, Richard Lewis, and Satinder
> > Singh. 2021. “Learning State Representations from Random Deep
> > Action-Conditional Predictions.” *arXiv:2102.04897*.
> > <http://arxiv.org/abs/2102.04897v1>.

---

> > > ### Comment · Reviewer_1S93 · 2021-11-23
> > > **My current take**
> > >
> > > My current take on the work is that it aims to do something interesting -- understanding how different types of generalization operate in RL settings but does not achieve this goal (I agree this is a big goal, and no single paper can achieve it, but this submission comes across as a bunch of disconnected thoughts).
> > >
> > > - The fact that the authors are not willing to add intuition, itself indicates to me that the study is incomplete.
> > >
> > > - There are too many things, with no direct connection, clubbed together and these don't lead to a coherent story. For example, I have gone over the paper twice now -- once at the submission time and once again just before writing this, and I can't still say what the takeaways are. Certainly, the takeaways can't be generalized from a few experiments, but that's a limitation of this work, since several prior works aiming at experimental analysis in RL do actually make at least a viable set of takeaways by experimenting on several domains and making conclusions via rigorous experiments.
> > >
> > > - I still feel like the paper touches on important questions, but only at a surface level, and with no clear further experiments (e.g, offline RL Atari experiments that authors say can be done), I cannot surely say that the paper has the impact and clear conclusions.
> > >
> > > - A big chunk of the takeaway is on seeing that prior experimental study conclusions do hold in their environment, but that cannot be a contribution of the paper for which the paper must be accepted here.
> > >
> > > To clarify, I am fine with gridworld only experiments (or only simpler domains) and I know Atari takes too long to run and is not representative of all RL problems, but enough rigorous experimentation for intuitions and takeaways seems somewhat necessary to me.
> > >
> > > Since the authors can respond in this week too, I am happy to wait for a reply before taking the final call on the score. If you could answer these points (primarily around intuitions, takeaways, and rigor) I am happy to reconsider my decision.

---

> > > > ### Author Response · Authors · 2021-11-29
> > > > **Greatly appreciate your extended discussion and engagement during the review process!**
> > > >
> > > >
> > > > Thanks for summarizing your current state. I understand that you feel
> > > > the ideas in the paper are disconnected.
> > > >
> > > > The central thesis is that, by
> > > > using contextual decision process, we are able to study generalization
> > > > in a single environment. Using this formalism, we go further to
> > > > disentangle the study of generalization across states, observations and
> > > > actions. The current experimental methodology and environments do not
> > > > allow this. Introducing observations is key because it allows us to
> > > > query an agent's action-value function with different inputs (train or
> > > > test) at the same state. This also allows the study of
> > > > state-generalization and action-generalization because the agent
> > > > leverages the regularity in the inputs (observations) to generalize to
> > > > either unseen states or unseen actions. This is not enough, however,
> > > > because we do not have anything to measure against the estimated
> > > > action-value. As shown in the paper (Equation 1 and footnote 2), value
> > > > can be written as an expectation of the scaled reward over the state
> > > > distribution. In Equation 2, we expand the set of criteria studied and
> > > > use dynamic programming on the underlying MDP to obtain ground-truth
> > > > estimates for the general criteria set. We evaluate several previous
> > > > observation generalization claims in the literature to first demonstrate
> > > > that our environment setup is reflective of other more commonly used
> > > > environments. We then show the nuances of these generalization
> > > > mechanisms by showing that observational findings do not necessarily
> > > > extrapolate to state and action generalization.
> > > >
> > > > As for the specific points you brought up:
> > > >
> > > > -   We chose to not include speculative intuition so as to not distract
> > > >     from our objective findings. We acknowledge, however, that intuition
> > > >     helps identify future work and helps contextualize the presented
> > > >     findings. In a future version, we will expand on the intuition that
> > > >     we discussed in the reviews.
> > > >
> > > > -   We acknowledge that our work does not easily fit into the current
> > > >     literature which makes the story seem less coherent. In a future
> > > >     version, we are including a motivating example that demonstrates the
> > > >     need for our characterization.
> > > >
> > > > -   There are many directions to take this work. We have focused on
> > > >     off-policy model-free value-based methods, but there are many
> > > >     sub-fields that hope to improve RL generalization or may benefit
> > > >     from studying their methods' effects on generalization: model-based
> > > >     RL, continual learning, auxiliary tasks, causal RL, intrinsic
> > > >     motivation (and other sophisticated exploration methods), curriculum
> > > >     learning, etc. There are many opportunities to clarify our
> > > >     understanding of generalization in RL.
> > > >
> > > >     Note, however, that the Atari experiment described in another reply
> > > >     only replicates the corruption experiment. Replicating our full
> > > >     experiment suite in offline Atari would require us to know the
> > > >     ground-truth values for the states in the testing replay buffer. But
> > > >     that is the cost of understanding generalization across states,
> > > >     observations and actions.
> > > >
> > > > -   This is not completely true, we focused on this in our updated draft
> > > >     to address a concern that our environment is not realistic. We argue
> > > >     it is realistic because it reproduces many findings in the
> > > >     literature. It is important to know that observation generalization
> > > >     findings can be reproduced, because it lends credence to our other
> > > >     findings in, not only observation generalization, but also state and
> > > >     action generalization.
> > > >
> > > > We greatly appreciate your extended discussion and engagement during the
> > > > review process!

---

> > ### Comment · Reviewer_1S93 · 2021-11-23
> > **Clarifications**
> >
> > Regarding the ground truth SL oracle: I didn't mean running supervised learning, but rather coming up with an optimal solution to the problem, for example, find $Q^*$ via some exact dynamic programming procedure, assuming knowledge of the environment, etc. Then regress to these solutions in a supervised fashion and see how well those networks generalize. Then we compare the generalization between this oracle obtained via exact dynamic programming (this is exact) but then supervised regression and the one obtained via RL algorithms. This allows us to compare the differences in generalization between RL and SL.
> >
> > For instance, imagine you could compute Q* via tabular value iteration, and then regress to it by training a network to predict Q* via supervised regression. You also trained Q via a DQN method. Now compare how these generalize. DQN vs QR-DQN is interesting, and there are many axes there to explore, but the supervised oracle may be a better point of comparison.

---

> > > ### Author Response · Authors · 2021-11-24
> > > **Thanks for the clarification**
> > >
> > > Thanks for the clarification! Yes, this can easily be done. The shortcoming of this experiment is that it is tangential to the problem we study, which is the generalization problem faced by RL agents. Whether a supervised learning approach can generalize better or worse than RL in our problem instance does not lead to any conclusions about the merit of an RL algorithm because no RL algorithm could implement this supervised learning approach. Still, we will consider your suggested experiment and devise a suitable hypothesis.

---

### Official Review · Reviewer_NqjA · 2021-11-03

**Correctness:** 3
**Technical Novelty And Significance:** 2
**Empirical Novelty And Significance:** 2
**Recommendation:** 5
**Confidence:** 4

**Main Review:**

I think the ideas in this paper are presented clearly and it is globally well-written. I like the distinction the authors made to distinguish generalization across their three axis is interesting and although they have been looked at separately in the literature, their juxtaposition and evaluation procedure in these different regimes are as far as I know novel. It also enables to measure generalisation in a single task setting and not across MDPs unlike some other works.

Originally, in statistical learning theory [Vapnik, 1995], the training and test distributions are the same in the definition of the generalisation gap (train error - test error) and the distribution only differs on the test in transfer learning so it would be good if you added a reference for your definition or precise why you are using a different definition.

Do you have some intuitions about why DQN and QR-DQN generalise similarly in the online setting on the observation space? or an hypothesis why DQN generalise better than qr-dqn on the space axis?

I think the empirical measure of an agent generalization capacity would be more convincing and realistic on more complex environment like some from the Atari game for instance. An empirical evaluation on more environment would also strengthen the paper. It would be interesting to also have results for other agents.

“While generalization of state-value is similar to regression, generalization with quantities related to action, such as policy or action-value, do not have supervised learning analogues and hence require separate consideration. “ could you add more justification for this please?

Here are a few minor points that did not affect my rating. Please use the proceedings links in your references instead of the arXiv links.

**Summary Of The Paper:**

This paper proposes an empirical evaluation method to measure of the generalization capacity of an RL agent. It relies on CDPs combining a tabular environment with a supervised learning dataset. Generalization is measured across three axis: state space, observation space and action space. The empirical evaluation is led on DQN and QR-DQN on the four room domain and corridor domain combined with the MNIST dataset in the online and offline settings. The authors find that dropout improves action generalization but not observation generalization while regularisation improves observation generalization. They also find that QR-DQN generalise better than DQN in the offline setting on the observation axis and action axis but not on the state space axis.

**Summary Of The Review:**

This work provides interesting insights comparing the generalization capacities of DQN and QR-DQN and their evaluation method corroborates some findings in the literature (e.g. about regularization) and some somewhat surprising results on the state generalization capacities of DQN vs QR-DQN. I think the paper would be stronger by providing results for other agents and more complex environments which is why I recommend a weak reject.

---

> ### Author Response · Authors · 2021-11-21
> **Response**
>
> Thank you for your review! Below we address the points that you raised:
>
> Difference in test and train distribution
> -
>
> You are correct that the distributions between testing and training are
> the same, in the population sense. The definition we provide is
> "operationalized" because we, as practitioners, partition the population
> into testing and training distributions for measurement.
>
> Intuitions
> -
>
> As discussed in the shared reply, we did not think it correct to include
> intuition on the nature of our results without room for further
> experimentation or connection to already established literature. Below,
> we speculate on our findings.
>
> Between L2 regularization and dropout, we find the surprising fact that
> dropout provides action generalization benefits and L2 regularization
> helps observation generalization. This can be explained by the fact that
> L2 penalizes weights globally across all inputs, which favors smoother
> solutions between observations/states. This explains the improvement of
> observation generalization across all experiments. Even though the
> training performance on unseen states (Figure 6, top middle) is lower
> with L2 regularization, the smoothness of the solution still improves
> the generalization gap (Figure 6, bottom middle). Dropout, on the other
> hand, provides local regularization for each input by turning off random
> weights. To compensate, the outputs in action space must be smoother but
> this smoothness does not occur between inputs.
>
> Between QR-DQN and DQN, we see in Figure 3 that QR-DQN can struggle to
> fit the corrupted training data as the number of heads increases. This
> means that, although QR-DQN has more capacity because of the number of
> heads, this capacity must be spent to predict more quantiles of the
> return distribution. The heads provide an inductive bias that helps
> generalization when data is limited, as in the offline regime (Figure 1,
> left). These heads can overfit however, as evidenced by the relatively
> worse action-generalization (Figure 4, left). This suggests that, in the
> online regime (Figure 2, left), QR-DQN is not able to use the
> distributional inductive bias to improve generalization because the
> heads are constantly changing their estimates of the training
> distribution for return as new transitions are encountered.
>
> Generalization capacity on Atari, other environments and agents
> -
>
> The capacity experiment could be conducted in an offline Atari
> experiment by shuffling all the observations in a replay buffer. This
> experiment would not be as helpful to elucidate the performance
> differences in our experiments however, because the architecture would
> differ for Atari. An online Atari capacity experiment is also not
> possible because it would require us to know the function mapping from
> Atari RAM state to pixel observations, which is unavailable.
>
> Experiments on other environments can be conducted, but they must be
> part of the contextual decision process family of environments described
> in our submission. Does the reviewer have specific suggestions on how
> our currently chosen MDPs can be improved to study the generalization
> across states, observations or actions?
>
> For other agents, we made the deliberate decision to focus on mechanisms
> that have been already established or claimed to improve generalization
> (distributional, L2, dropout). With this restriction, there are not many
> claims outside of the observation that QR-DQN does better in the offline
> setting (Agarwal, Schuurmans, and Norouzi 2020). One general family of
> algorithms that we would like to investigate is model-based RL
> algorithms. This addition is out of the scope of our current paper
> however, where we first focus and clarify the generalization of
> model-free agents.
>
> Justification regarding action-related quantities in RL
> -
>
> Thank you, this is a subtle point that deserves clarification. When
> learning the state-value with Monte-Carlo methods, we are regressing the
> state onto the return. This has a clear analogy to supervised learning.
> When introducing action-values, this analogy is less accurate even for
> Monte-Carlo action-value learning. The reason is that the regression
> function now has an output for every action, and there is a shared
> representation between the actions even if a single "action head" is
> being updated at a time. The same is true for policy optimization, where
> we do not have true "labels" for the optimal action but local
> approximations based on the estimated value or a Monte-Carlo rollout for
> a specific action. This is why the introduction of actions requires
> special consideration and, as we argue, its own measure of
> generalization.
>
> References
> -
>
> Agarwal, Rishabh, Dale Schuurmans, and Mohammad Norouzi. 2020. “An
> Optimistic Perspective on Offline Reinforcement Learning.” In
> *Proceedings of the 37th International Conference on Machine Learning,
> ICML 2020, 13-18 July 2020, Virtual Event*, 119:104–14. Proceedings of
> Machine Learning Research. PMLR.
> <http://proceedings.mlr.press/v119/agarwal20c.html>.

---

> > ### Comment · Reviewer_1S93 · 2021-11-23
> > **Some questions on author response**
> >
> > Thanks for your reply! While this is not my review, I am still not sure about the bit relating to action-related quantities. In my opinion, it is incorrect to say that state-values generalize similar to regression and action-values don't. It is not just a function of having action heads or not (and in continuous-action domains, we will have s, a, both as inputs), but rather a function of the bootstrapping update vs the supervised learning update.
> >
> > Measuring action generalization separately is fine with me, I don't buy the explanation you gave above.

---

> > > ### Author Response · Authors · 2021-11-24
> > > **Re: continuous state/action action-values and bootstrapping**
> > >
> > > That is an excellent point about continuous state and actions. When the
> > >     state and action are both inputs for an action-value function, then the
> > >     action input has a direct effect on the activations of the neural network
> > >     and on the single scalar action-value output. In this case,
> > >     generalization in action-space can be treated similarly to state (or
> > >     observation) generalization for its respective input.
> > >
> > > Restricting our attention to discrete actions: you are correct that
> > >     bootstrapping is also an issue, but this is a separate issue from what we are highlighting. Our reply
> > >     mentioned value estimation using Monte-Carlo returns to highlight the fact
> > >     that different action heads at the output level require special
> > >     consideration as far as generalization is concerned. Learning action-values
> > >     using Monte-Carlo returns in the discrete action setting involves only one
> > >     action-head, corresponding to the action taken (i.e. fitting
> > >     $(q_\theta(S_t,A_t) - G_t)^2$, where the action indexes the output $q_\theta(S_t,A_t) = q_\theta(S_t)[A_t] $). This is unlike your example where $A_t$ is an input and has learnable weights associated with it.
> > > Edit: The same argument follows for policy gradient methods. The gradient associated with REINFORCE is $\mathbb{E}_\pi \left[ G_t \frac{\nabla \pi (A_t | S_t, \theta}{\pi(A_t | S_t, \theta)}\right]$. In this case, the policy is only being updated for the specific action $A_t$ weighted by the Monte-Carlo return for taking action $A_t$ in state $S_t$ and following the policy $\pi$.

---

### Author Response · Authors · 2021-11-09
**Shared response to reviewers, individual replies in the coming days**

Thank you for your thoughtful reviews. We would like to first clarify
the contributions of this work:

1.  We provide a problem formulation for studying generalization in RL
    using contextual decision processes. While the contextual decision
    process is a previously established theoretical setting, it has not
    been used to study generalization in RL.

2.  Using this problem formulation, we disentangle the study of
    generalization in RL across states, observations and actions. This
    is a natural characterization that has not been previously studied
    because popular environments and benchmarks do not provide
    ground-truth values and policies while also providing rich
    observations for generalization.

3.  Next, we study existing generalization claims in the literature with
    two environments provided by our problem setting with different
    structure in the states, observations and actions. In the
    observational generalization setting, we: validate QR-DQN's
    superiority in the offline regime (Agarwal, Schuurmans, and
    Norouzi 2020), validate the benefit of small amounts of L2
    regularization across both online and offline regimes (Cobbe et
    al. 2019), and refute the benefits of dropout (Farebrother, Machado,
    and Bowling 2018).

4.  Lastly, we provide a preliminary investigation of generalization in
    state and action-space. We show that, while unhelpful for state and
    observation generalization, dropout does in fact help action
    generalization. We also find that QR-DQN's superiority to generalize
    to new observations in the offline regime does not translate to
    state-space generalization, where DQN is superior. This is the first
    study that shows substantial differences in state and action
    generalization compared to observation generalization.

Most reviewers considered the empirical section to be the only
contribution, but this is not the case. In the literature, both the
evaluation and the characterization of generalization in RL is lacking.
Our paper is a first step forwards to rectifying both of these issues.
The current methodology for evaluating generalization is more akin to
transfer learning, and does not provide enough experimental acuity to
properly measure generalization. Our goal is not a new environment or
algorithm, but an understanding of what generalization is in RL, and how
to properly measure it.

The reviewers also feel that similar results should be shown on
environments like Atari. The same experiment setup is not possible in Atari because the ground-truth
values, and the partitioning of states/observations/actions, provided by our problem formulation are necessary to study
generalization. I understand that there is some friction in using a
non-standard environment, but this is one of the central contributions
of our work. We argue that relatively simple environments are critical
to our understanding of the generalization properties of RL algorithms:
they enable us to make controlled and targeted experiments to validate
or refute generalization claims in the literature. Our setting already
allows us to probe the generalization limits of modern deep RL
algorithms. Significant differences arise in this minimal
setting. This is evidenced by the test and train discrepancies of DQN
and QR-DQN across all experiments, despite having the same representational
architecture and, in the offline regime, using the same data.

As for any intuition for why the results are the way they are, we have a
number of hypotheses that we would like to test. We would like to remind
the reviewers, however, that this paper is already very divergent from
the current state of RL generalization research. While we plan to
discuss some of these hypotheses and intuition in the individual
replies, we did not think it appropriate to include in the main paper if
there was no room for further validation experiments.

In the coming days, we will be uploading comments addressing specific
points made by each reviewer.

Agarwal, Rishabh, Dale Schuurmans, and Mohammad Norouzi. 2020. “An
Optimistic Perspective on Offline Reinforcement Learning.” In
*Proceedings of the 37th International Conference on Machine Learning,
ICML 2020, 13-18 July 2020, Virtual Event*, 119:104–14. Proceedings of
Machine Learning Research. PMLR.
<http://proceedings.mlr.press/v119/agarwal20c.html>.

Cobbe, Karl, Oleg Klimov, Christopher Hesse, Taehoon Kim, and John
Schulman. 2019. “Quantifying Generalization in Reinforcement Learning.”
In *Proceedings of the 36th International Conference on Machine
Learning, ICML 2019, 9-15 June 2019, Long Beach, California, USA*,
edited by Kamalika Chaudhuri and Ruslan Salakhutdinov, 97:1282–9.
Proceedings of Machine Learning Research. PMLR.
<http://proceedings.mlr.press/v97/cobbe19a.html>.

Farebrother, Jesse, Marlos C. Machado, and Michael Bowling. 2018.
“Generalization and Regularization in Dqn.” *arXiv:1810.00123*.
<http://arxiv.org/abs/1810.00123v3>.

---

### Author Response · Authors · 2021-11-23
**Updated draft, main concerns summarized and addressed below**

At the time of posting, only 1 reviewer has responded to either our shared or
individual rebuttal. I would like to remind the reviewers to let us know
if their concerns are addressed.

We have uploaded a revision of our paper that addresses the concerns
shared amongst the reviewers. Changes in the manuscript are highlighted
in blue. Below, we summarize the overall concerns shared by the
reviewers and how we address them:

Overall significance and novelty
--------------------------------

Our paper introduces a new characterization of generalization in
reinforcement learning within a single environment and provides a novel
empirical study investigating this characterization. Our perspective on
generalization is significant because it is uniquely analogous to
generalization in supervised learning. This perspective allows us to
disentangle generalization across states, observations and actions. We
validate several common findings concerning generalization in RL, while
also providing new insights on how different generalization mechanisms
benefit different generalization axes.

We have made some changes to the introduction to better position our
submission.

Summarizing takeaways and an argument for why they would generalize to other environments
--------------------------------------------------------------------

We have added Subsection 4.4 to discuss the empirical findings based on
the reviews, our rebuttals and the response from Reviewer iELT. We
discuss key takeaways and the reasons why these results can be expected
to generalize to environments where our experimental study cannot be
conducted.

Importance of criteria other than sum of reward
-----------------------------------------------

Some reviewers have expressed skepticism about why the generalization
criteria introduced should be of interest. Recall that the definition of
value can be written as an expectation over a stationary state
distribution \\(d_s^\pi\\), where
\\(g(s', o', a', \pi) = -\frac{1}{1 - \gamma}r(s', a')\\). This is
equivalent to value (i.e. the expected sum of reward), but fails to
disentangle generalization, unlike our general formulation, because it
is trajectory dependent - \\(\pi\\) appears in the action selection and
the state distribution, but not in \\(g\\) itself - which conflates
generalization across states, observations and actions. Our more general
criteria can separate generalization across states, observations and
actions, allowing us to understand when certain algorithms or
generalization mechanisms improve one axis rather than another.

We have added further clarification in Section 3 to
provide better motivation for the alternative generalization criteria that we
investigate in our experiments.

Other changes in light of reviewer comments
-------------------------------------------

-   Clarified connection between our work and related work in
    introduction.
-   Justified difference of action-related quantities.
-   Clarified training and testing set come from the same population distribution in Equation 2.
-   Added motivation for experiment setup.
-   References to papers on arXiv have been replaced with references to
    peer-reviewed venues where available.
-   Added explanation for how results are visualized in Appendix.
-   The incorrect Figure labels have been fixed.
-   Some typos have been fixed.
-   An illustration of the environments has been included in the
    Appendix.
-   Updated Dropbox code with instructions to generate figures for
    different selection criteria (best, final or AUC during training).
    As well as controlling for particular hyperparameter settings.

---

### Decision · Program_Chairs · 2022-01-20

**Decision:**

Reject

**Comment:**

This work proposes to study the generalization capabilities of RL algorithms using contextual decision processes (CDPs). CDPs allows to study generalization similar to how we are used to studying generalization in supervised learning, and can separate the generalization capabilities of a learned agent wrt observation, state and action space. This proposed measure for generalization is used in an extensive study on grid world domains to evaluate existing algorithms that aim to improve generalization.

**Strengths**
This manuscript is well written and the work is well motivated
A novel perspective and way of measuring generalization of learned agents
An empirical study that compares existing algorithms on how well they generalize in observation, state, action spaces

**Weaknesses**
Some clarity issues existed (missing links to existing literature, experimental details)
empirical study is (out of necessity) limited to small scale grid worlds
no deeper analysis of the results, why do algorithms perform the way they do from this novel perspective of generalization, which makes it hard to understand how one could choose an algorithm for larger scale settings which don't allow for this type of analysis

**Rebuttal**
The authors updated the paper to improve the parts that were unclear, and had an extensive discussion with reviewers on the intuition of the results and converging on take-aways. Unfortunately, this intuition and take-aways have not been added.

**Summary**
While I understand the authors wish to not speculate on intuition, I agree with the reviewers that without (experimentally supported) take-aways the provided analysis is incomplete. Understanding why each algorithm achieves the performance they do wrt this novel way of measuring generalization is the only way the proposed method to measure generalization and the evaluation can be used to draw conclusions about more general problem settings. Thus, although this is a very promising direction on an important problem, the manuscript is not ready yet for publication.